# Gemcitabine-Vitamin E Prodrug-Loaded Micelles for Pancreatic Cancer Therapy

**DOI:** 10.3390/pharmaceutics16010095

**Published:** 2024-01-10

**Authors:** Miguel Pereira-Silva, Darío Miranda-Pastoriza, Luis Diaz-Gomez, Eddy Sotelo, Ana Cláudia Paiva-Santos, Francisco Veiga, Angel Concheiro, Carmen Alvarez-Lorenzo

**Affiliations:** 1Department of Pharmaceutical Technology, Faculty of Pharmacy of the University of Coimbra, University of Coimbra, 3000-548 Coimbra, Portugal; miguelsp_20@hotmail.com (M.P.-S.); acsantos@ff.uc.pt (A.C.P.-S.); fveiga@ci.uc.pt (F.V.); 2REQUIMTE/LAQV, Group of Pharmaceutical Technology, Faculty of Pharmacy of the University of Coimbra, University of Coimbra, 3000-548 Coimbra, Portugal; 3Departamento de Farmacología, Farmacia y Tecnología Farmacéutica, I+D Farma (GI-1645), Facultad de Farmacia, Instituto de Materiales (iMATUS) and Health Research Institute of Santiago de Compostela (IDIS), Universidade de Santiago de Compostela, 15782 Santiago de Compostela, Spain; luis.diaz.gomez@usc.es; 4Department of Organic Chemistry, Faculty of Farmacy, Universidade de Santiago de Compostela, 15782 Santiago de Compostela, Spain; dario.miranda.pastoriza@usc.es (D.M.-P.); e.sotelo@usc.es (E.S.); 5Centro Singular de Investigación en Química Biolóxica e Materiais Moleculares (CIQUS), Universidade de Santiago de Compostela, 15782 Santiago de Compostela, Spain

**Keywords:** polymeric micelle, pancreatic cancer, chemotherapy, gemcitabine, drug delivery, vitamin E conjugate, nanoparticle, prodrug, excipient reduction

## Abstract

Pancreatic cancer (PC) is an aggressive cancer subtype presenting unmet clinical challenges. Conventional chemotherapy, which includes antimetabolite gemcitabine (GEM), is seriously undermined by a short half-life, its lack of targeting ability, and systemic toxicity. GEM incorporation in self-assembled nanosystems is still underexplored due to GEM’s hydrophilicity which hinders efficient encapsulation. We hypothesized that vitamin E succinate–GEM prodrug (VES-GEM conjugate) combines hydrophobicity and multifunctionalities that can facilitate the development of Pluronic^®^ F68 and Pluronic^®^ F127 micelle-based nanocarriers, improving the therapeutic potential of GEM. Pluronic^®^ F68/VES-GEM and Pluronic^®^ F127/VES-GEM micelles covering a wide range of molar ratios were prepared by solvent evaporation applying different purification methods, and characterized regarding size, charge, polydispersity index, morphology, and encapsulation. Moreover, the effect of sonication and ultrasonication and the influence of a co-surfactant were explored together with drug release, stability, blood compatibility, efficacy against tumour cells, and cell uptake. The VES-GEM conjugate-loaded micelles showed acceptable size and high encapsulation efficiency (>95%) following an excipient reduction rationale. Pluronic^®^ F127/VES-GEM micelles evidenced a superior VES-GEM release profile (cumulative release > 50%, pH = 7.4), stability, cell growth inhibition (<50% cell viability for 100 µM VES-GEM), blood compatibility, and extensive cell internalization, and therefore represent a promising approach to leveraging the efficacy and safety of GEM for PC-targeted therapies.

## 1. Introduction

Pancreatic cancer (PC) represents one of the deadliest known cancers worldwide, accounting for a dismal 5-year survival rate of only 9% [1,2]. Exocrine pancreas-originated pancreatic ductal adenocarcinoma (PDAC) accounts for ca. 90% of all PC cases and is characterized by its aggressiveness and high metastatic profile, as well as late detection rates and a lack of efficient therapeutics [1]. The complex PDAC pathological frameworks include a desmoplastic stroma barrier surrounding PDAC cells [3], composed of cancer-associated fibroblasts, a collagen-enriched extracellular matrix, and a low fraction of immune cells. The high hydrostatic pression in the tumour limits drug penetration, which is worsened by the emergence of multidrug resistance (MDR), as PDAC cells rapidly develop resistance against conventional chemotherapeutics [4]. Moreover, there is an immunosuppressive tumour microenvironment characterized by a reduced immune cell infiltrate and immune system suppression blocking immune-mediated tumour cell eradication [5].

Chemotherapy remains one of the main therapeutic approaches for PC management, together with surgical removal of the tumour and radiotherapy. Gemcitabine (2′,2′-difluoro-2′-deoxycytidine; GEM) is an antimetabolite compound, a nucleoside analogue (deoxycytidine analogue) that acts in two main ways: by inhibiting the synthesis of DNA and cell growth, and also through self-potentiation, by inhibiting the enzymes relevant for nucleotide metabolism [2,4]. Gemcitabine, which is a biopharmaceutical classification system (BCS) class III drug (high water solubility, low permeability), enters cells through nucleoside transporters (hNTs) and requires further phosphorylation steps for activation. Specifically, GEM is activated intracellularly by deoxycytidine kinase and phosphorylated into active forms. GEM is rapidly deactivated by the cytidine deaminase present in several tissues to 2′,2′-difluoro-2′-deoxyuridine (dFdU) and eliminated by the body, evidencing a short blood circulation half-life [6]. Despite its widespread use, conventional chemotherapy is known to have limited effectiveness, and its intravenous administration shows serious challenges, such as low serum stability, severe systemic toxicity, and the rapid emergence of an MDR scenario. GEM presents only limited efficacy owing to the PDAC pathological profile, associated not only with the complex environment of this tumour but also with the low capability of hydrophilic GEM to enter cells via diffusion and its rapid clearance, which seriously undermine PDAC therapeutics [7,8]. GEM encapsulation in nanosystems may, in part, successfully address these drawbacks by improving serum stability and blood circulation half-life as well as minimizing systemic toxicity and facilitating tumour biodistribution and cell penetration [8,9]. Stability and blood circulation improvements may be attributed to the stealth features of PEGylated nanosystems which can help in maximizing the blood circulation profile of drugs and decrease aggregation [10]. Additional stimuli-responsive attributes and advanced targeting moieties can be included in nanosystems to improve tumour targeting and enable selective drug release, in accordance with natural and pathophysiological barriers [11].

Polymeric micelles (core–shell self-assembled amphiphilic polymers) are suitable for PDAC-targeted therapies owing to their tailorable size, shape, and composition properties, easily surface-functionalized surface, and ability to incorporate hydrophobic drugs into their cores [12]. In contrast to the majority of chemotherapy drugs, GEM is highly hydrophilic and its encapsulation in the hydrophobic core of micelles is not directly feasible [13]. Conjugating GEM to lipophilic compounds may overcome the low efficiency of its encapsulation into micelles while also facilitating penetration into tumour cells [14,15]. To date, several lipidic blocks have been reported in the literature as being able to assemble GEM–lipid conjugates, namely N-octanoyl [16], N-dodecanoyl [17], 4-N-stearoyl (C_18_) [18], stearic acid [18,19,20,21,22,23,24], linoleic acid [25], cholesterol [26,27], squalene [15,28,29,30,31,32,33,34], pentadecanoic acid [35], vitamin E (VE) [36], and VE derivatives such as VE succinate [37,38,39,40,41]. Some GEM–lipid conjugates are able to release GEM in the presence of internal stimuli, such as the lysosomal proteolytic enzyme cathepsin B [42,43], which is frequently overexpressed in several cancer types, phospholipases [44] and carboxylesterases [45]. Additionally, the stimuli-responsiveness properties of the lipid–GEM conjugates enable site-specific activation of the prodrug system, leading to GEM controlled release, and endowing these nanosystems with enhanced drug release properties as well as minimizing burst release. The incorporation of GEM prodrugs in polymeric micelles has been successfully reported; specifically, 4-(N)-stearoyl GEM (denoted as GEM-C_18_) in poly(ethylene glycol)–poly(d,l-lactide) (PEG–PLA) micelles [23] and PEG-distearoylphosphatidylethanolamine (PEG-DSPE)/tocopheryl polyethylene glycol 1000 succinate (TPGS) micelles [20]. These studies demonstrated an increase in GEM prodrug stability, blood circulation half-life, and tumour biodistribution [20].

Vitamin E-based or -containing nanosystems are receiving increasing attention as functional platforms for chemotherapy delivery, able to provide additional antioxidant and anticancer properties [46]. Particularly, vitamin E succinate (VES, Figure 1) is frequently used as building block for drug conjugates that exhibits synergistic anticancer activity. The biofunctional and structural 2 in 1 properties ascribed to VES constitute clear advantages for further employment of VES-based micelles as efficient and multifunctional nano-based platforms assembling VES-GEM prodrugs (Figure 1) for improved GEM delivery to PDAC.

Pluronic^®^ F68 and Pluronic^®^ F127 are both versatile nonionic amphiphilic triblock copolymers of poly(ethylene oxide) (PEO) and poly(propylene oxide) (PPO) (Figure 1) with remarkable self-assembly properties [47,48,49] already used in FDA-approved medicines, showing improved pharmacokinetics of hydrophobic drugs and MDR reversal, namely via P-glycoprotein (P-gp) inhibition [50]. PEO units can provide a hydrophilic shell with stealth properties, thus minimizing interactions with serum proteins and avoiding recognition by the reticuloendothelial system [24], and contributing to enhanced tumour biodistribution via the EPR effect, while hydrophobic PPO units enable drug encapsulation mainly through hydrophobic interactions [51]. In addition, Pluronic^®^ F68 and Pluronic^®^ F127 display physical state of flake (F), high hydrophilicity, solubility, and biocompatibility. Their hydrophilic–lipophilic balance (HLB) value is within 20–29 and the PPO chain length of Pluronic^®^ F127 is twice that of Pluronic^®^ F68 [48,49]. Recently, mixed Pluronic^®^ F68 micelles were loaded with docetaxel (20 mg/mL F68, 2% *w*/*v*; 0.8 mg/mL docetaxel, 0.08% *w*/*v*) [52] and curcumin (5 mg/mL F68, 0.5% *w*/*v*; 1 mg/mL curcumin, 0.1% *w*/*v*) [53] and Pluronic^®^ F127 (F127) micelles were investigated for delivery of curcumin (20 mg/mL of F127, 2% *w*/*v*; 0.1 mg/mL, 0.01% *w*/*v* curcumin) [54], alpha-lipoic acid (20 mg/mL of F127, 2% *w*/*v*; 10 mg/mL, 1% *w*/*v*) [55], hypericin [56], β-escin (5 mg/mL F127, 0.5% *w*/*v*, 5 mg/mL β-escin, 0.5% *w*/*v*) [57], boron-dipyrromethene dimer [58], lenvantinib [59], and a photosensitizer (1.11 mg/mL F127, 0.11% *w*/*v*) [60], hence comprising biofunctional and suitable delivery systems for loading poorly water-soluble drugs with enhanced stability. Mixed Pluronic^®^ micelles have also been reported in the literature as a means to increase the solubility profile of poorly water-soluble drugs [61,62]. On the other hand, Pluronic^®^ micelles loaded with GEM prodrugs have not yet been reported, to the best of authors’ knowledge.

The aim of this work was to develop multifunctional vitamin E succinate-gemcitabine (VES-GEM) conjugate-loaded Pluronic^®^ F68 and Pluronic^®^ 127 micelles (denoted as PF68/VES-GEM and PF127/VES-GEM) via a hydrophobic GEM prodrug derivative strategy to: (1) successfully encapsulate GEM in the hydrophobic core of the micelle; (2) increase the stability of GEM by preventing deamination at the 4-(N) position; (3) improve the lipophilicity of GEM, enabling cell entry through diffusion or endocytosis; (4) enable controlled and site-specific release under multivariate physiological stimuli, as well as acting as a GEM reservoir for ultra-long VES-GEM protection and release by partial VES-GEM release and increased affinity to the micelle core; and (5) follow an excipient-reduction rationale towards increased safety and drug loading. The effect of micelle composition on physicochemical parameters was firstly studied by preparing PF68/VES-GEM and PF127/VES-GEM micelles via a solvent evaporation method, bearing distinct molar ratios and characterized regarding size, surface charge and polydispersity index, morphology, encapsulation efficiency, and drug loading. The addition of a co-surfactant was also studied by preparing mixed micelles of Pluronic^®^ and an amphiphilic graft copolymer, polyvinyl caprolactam polyvinyl acetate-polyethylene glycol Soluplus^®^, assembling Pluronic^®^/Soluplus^®^@VES-GEM micelles, which were characterized as described above. The release profile was also assessed to ascertain whether the micelles could display controlled VES-GEM release, and the stability of the micelles was also evaluated. Lastly, in vitro studies using the PDAC cell line BxPC3 were performed to elucidate cell viability and cellular uptake of PF68/VES-GEM and PF127/VES-GEM micelles. Overall, a strategy was devised to enhance GEM stability and delivery to PDAC cells towards improved PDAC therapeutics by means of biofunctional Pluronic^®^ micelles.

## 2. Materials and Methods

### 2.1. Materials

The vitamin E succinate (VES; MW = 530.80 g/mol) was purchased from Santa Cruz Biotechnology Inc., Dallas, TX, USA; the Soluplus^®^ (polyvinyl caprolactam-polyvinyl acetate-polyethylene glycol copolymer) (MW~115,000 g/mol) was purchased from BASF^®^, Ludwigshafen, Germany; the sodium 1-heptanesulfonate (MW = 220.26 g/mol), was purchased from Fluka Chemie, Buchs, Switzerland; the triethylamine, anhydrous sodium sulphate (Na_2_SO_4_), and sodium bicarbonate (NaHCO_3_) were purchased from Sigma-Aldrich, St. Louis, MO, USA; the N,N-dimethylformamide (DMF), ethyl acetate (C_4_H_8_O_2_), and di-sodium hydrogen phosphate di-hydrate (Na_2_HPO_4_.2H_2_O), Millex^®^ syringe-driven filter unit and the water was purified using the Milli-Q^®^ (Millipak^®^ 0.22 μm) system, all obtained from Merck, Darmstadt, Germany; the tetrahydrofuran (THF) and dichloromethane (DCM), and gemcitabine hydrochloride (GEM·HCl, C_9_H_11_F_2_N_3_O_4_·HCl) (MW = 299.66 g/mol) were purchased from Fisher Scientific, Loughborough, UK; the isobutyl chloroformate was purchased from ThermoFisher GmBH, Kandel, Germany; the silica gel (60 Å, 35–70 µ) was purchased from Carlo Erba reagents, Val de Reuil,, France; the sodium hydroxide (NaOH) was purchased from VWR Chemicals, Leuven, Belgium; the sodium dihydrogen phosphate anhydrous (NaH_2_PO_4_) and sodium chloride (NaCl) were purchased from Labkem, Barcelona, Spain; the potassium chloride (KCl) and potassium di-hydrogen phosphate (KH_2_PO_4_) were obtained from Panreac Quimica S.L.U., Barcelona, Spain. The hydrophilic polytetrafluoroethylene (PTFE) syringe filter (13 mm, 0.2 µm) and hydrophilic PTFE syringe filter (25 mm, 0.4 µm) were purchased from Scharlab S.L., Barcelona, Spain. The syringe filter (30 mm, 0.45 µm) with a polyethersulfone membrane, sterile, was purchased from JET BIOFIL^®^, Huangpu District, Guangzhou, China. The nylon membrane discs (47 mm, 0.2 µm) were obtained from Pall Corporation, Waters, Ann Arbor, MI, USA. The buffered solution was phosphate-buffered saline (PBS, 0.01 M, 1 X pH = 7.4), namely 8 g/L NaCl, 0.2 g/L KCl, 1.44 g/L Na_2_HPO_4_.2H_2_O, 0.24 g/L KH_2_PO_4_. To prepare the phosphate-buffered saline (PBS, 0.01 M, 1 X pH = 5), the pH was adjusted by varying the concentration of buffer species: 8 g/L NaCl, 0.2 g/L KCl, 0.223 g/L Na_2_HPO_4_.2H_2_O, 13.39 g/L KH_2_PO_4_. All additional reagents were of analytical or HPLC grade and used following the manufacturer’s instructions.

### 2.2. Synthesis of the VES-GEM Conjugate

The synthesis protocol was adapted from a previous report [39] and run in triplicate. In brief, VES (0.53 g, 1 mmol) was dissolved in 5 mL of anhydrous THF, followed by TEA (0.17 mL, 1.2 mmol) addition to the stirred solution and cooling (−15 °C) [39]. Then, a solution of isobutyl chloroformate (0.13 mL, 1.2 mmol) in anhydrous THF (5 mL) was added to the reaction mixture by applying the dropwise addition method, followed by stirring (−15 °C for 30 min). Next, the GEM·HCl (0.300 g, 1 mmol) and TEA (0.17 mL, 1.2 mmol) were dissolved in anhydrous DMF (5 mL) and the obtained solution was added dropwise to the reaction mixture (−15 °C). After stirring at room temperature for 72 h, the reaction mixture was concentrated in vacuo using Rotavapor^®^ R-300 (Buchi, Flawil, Switzerland) to evaporate the majority of the solvent. The reaction mixture was then subjected to liquid–liquid extraction with ethyl acetate (3 mL/50 mL) in the presence of aqueous sodium hydrogen carbonate and washed with water, which was followed by drying over anhydrous Na_2_SO_4_. Then, the extracts were concentrated under reduced pressure using Rotavapor^®^ R-300 to evaporate all of the solvent and yield a white crystalline solid product, further purified by chromatography on silica gel eluting a solid product with 0.5–5% methanol in DCM through wetted silica gel. The corresponding solution of the conjugate was collected and evaporated Rotavapor^®^ R-300 (Buchi, Flawil, Switzerland) to give pure VES-GEM as a white crystalline solid, stored at 4 °C for further use.

### 2.3. Characterization of the VES-GEM Conjugate

The yield of the VES-GEM conjugate synthesis process was calculated using Equation (1):(1)$$\mu =\frac{Weighted\;amount\;of\;obtained\;VES-GEM\;conjugate\;(mg)}{Theoretical\;amount\;of\;VES-GEM\;(mg)}$$

The obtained conjugates were characterized through Fourier-transform infrared spectroscopy (FTIR) using JASCO FTIR-4100 model. Proton nuclear magnetic resonance (^1^H NMR) and fluorine-19 nuclear magnetic resonance (^19^F NMR) spectra were recorded on Bruker AM300 and XM500 spectrometers using DMSO-d6 as the solvent. Chemical shifts were given as d values against tetramethylsilane as internal standard and J values were given in Hertz (Hz). Mass spectroscopy (MS) was conducted to obtaining mass spectra using an Autospec Micromass spectrometer (Waters, Milford, MA, USA). ChemDraw software, version 20.1.1 (PerkinElmer, Waltham, MA, USA) was used to determine the pKa, LogS, and LogP of the VES-GEM conjugate. Chemicalize software and MarvinView software (http://chemaxon.com, ChemAxon Ltd., Budapest, Hungary) were used to assess the 2D and 3D structure of the conjugate, the molecular weight, isotope formula, isoelectric point, LogP, LogD, HLB, intrinsic solubility, solubility at pH 7.4, predominant species (pH 4–9 range) and charge (pH 4–9 range), 3D conformation, and hydrogen bond donor and acceptor sites, and the predicted ^1^H NMR and ^13^C NMR spectra were also collected.

### 2.4. HPLC Quantification Methods

A RP-HPLC system JASCO (Tokyo, Japan), equipped with AS-4150 RHPLC Autosampler, PU-4180 RHPLC Pump, LC-NetII/ADC Interface Box, CO-4060 Column Oven, MD-4010 Photo Diode Array Detector, and ChromNAV software, version 2.0 (JASCO, Tokyo, Japan) was used. For the quantification of the VES-GEM conjugate, a Zorbax Eclipse XDB-C18 column (5 μm, 4.6 mm × 250 mm, Agilent Technologies, Santa Clara, CA, USA) at 30 °C and methanol as the mobile phase, with a flow rate 1 mL/min, was used. The detection wavelength was set at 248 nm [39]. All samples were filtered before injection through a PTFE hydrophilic Scharlau syringe filter (13 mm, 0.22 µm). The injection volume was 20 µL. For the VES-GEM calibration curve, a stock solution of VES-GEM in ethanol (50 ppm, 10 mL) was progressively diluted to standard solution concentrations of 40 ppm, 30 ppm, 20 ppm, 10 ppm, 5 ppm, and 1 ppm. The calibration curve of the quantified VES-GEM concentration, as a function of peak area (absorbance detected) (Appendix A), was drawn and showed good linearity within a concentration range of 1–50 ppm: linear regression equation y = 20779x + 2003.5, R = 0.9995, LOD = 0.15 ppm, LOQ = 0.236 ppm. The retention time was t = 8.0 min (Appendix A).

### 2.5. Stability of GEM and the VES-GEM Conjugate

The stability of GEM and the VES-GEM conjugates at 4 °C and 37 °C was assessed by measuring their content using HPLC. Briefly, stock solutions of GEM in water (200 ppm) and VES-GEM in ethanol (200 ppm) were prepared and placed in vials left at 4 °C in the fridge and incubated at 37 °C in an incubation shaker (Incubator 1000, Heidolph Instruments GmbH & Co. KG, Schwabach, Germany), for eight and four weeks, respectively. Aliquots were collected at predetermined time points, and both the free GEM and VES-GEM contents were measured by HPLC in an attempt to test the variation in concentration of each stock solution through time. The stability of both the free GEM and VES-GEM conjugate at acidic conditions typical of the endolysosomal compartment (pH = 5) was also assessed following a previous procedure, only this time pH was adjusted to 5 using HCl (1 M) and the vials were kept at 4 °C and 37 °C for the VES-GEM conjugate, and, for free GEM, both HCl (1 M) and PBS pH = 5 medium were used to achieve acidic conditions, and the vials were kept at RT. The photostability assay was carried out using a UV irradiation chamber, I42 (Heraeus Noblelight GmbH, Hanau, Germany), followed by HPLC quantification. Briefly, the vials containing the GEM and VES-GEM solutions (200 ppm) were placed in a UV irradiation chamber for 24 h, and aliquots were collected at t = 0 h, t = 1 h, and t = 24 h and the respective contents analysed through HPLC. In order to test the stability of the stored VES-GEM conjugate, ^1^H NMR was routinely performed to ensure that the VES-GEM conjugate had maintained stability.

### 2.6. Pluronic^®^/VES-GEM Micelle Preparation

A set of Pluronic^®^ F68/VES-GEM and Pluronic^®^ F127/VES-GEM micelles with a varying polymer-to-conjugate molar ratio were prepared using the solvent evaporation method (Table 1 and Figure 2) [63]. The VES-GEM conjugate concentration was fixed and the Pluronic^®^ concentration was varied. Briefly, a set of solutions of the VES-GEM conjugate (4 mg) in ethanol (5 mL) were prepared by pouring ethanol into glass vials containing the weighted amount of conjugate and stirring at 600 rpm (Cimarec i Poly 15; Thermo Scientific™, Fisher Scientific S.L., Madrid, Spain) at room temperature for 3 h until complete dissolution. Then, different amounts of Pluronic^®^ F68 (32.26 mg, 64.52 mg, 129.04 mg, 258.08 mg, corresponding to a 0.75/1, 1.5/1, 3/1, and 6/1 Pluronic^®^ F68-to-conjugate molar ratio, respectively) or Pluronic^®^ F127 (48.39 mg, 96.78 mg, 193.56 mg, corresponding to a 0.75/1, 1.5/1, and 3/1 Pluronic^®^ F127-to-conjugate molar ratio, respectively) were individually added to PBS diluted in water (50:50 *v*/*v*, 15 mL), which was followed by stirring (300 rpm) at room temperature for 3 h. The obtained VES-GEM conjugate solutions were then added dropwise to polymer solutions under gentle stirring and kept under magnetic stirring and protected from light at RT overnight (14–16 h) to enable solvent evaporation and Pluronic^®^/VES-GEM micelle formation. Pristine Pluronic^®^ F68 and Pluronic^®^ F127 micelles were prepared accordingly by dissolving the required amount of Pluronic^®^ F68 and Pluronic^®^ F127 in PBS diluted in water (50:50 *v*/*v*, 15 mL).

### 2.7. Influence of the VES-GEM Conjugate on the Micellization Process

To evaluate the influence of the VES-GEM conjugate on the self-assembly of Pluronic F68^®^ (CMC 0.48 mM) and Pluronic^®^ F127 (CMC 0.0028 mM) micelles, a surface tension assay was carried out [64]. Briefly, two stock solutions of VES-GEM in ethanol (500 µL, 37.24 mg/mL, and 500 µL; 2.173 × 10^−1^ mg/mL) were prepared, for a CMC assay of Pluronic F68^®^ and Pluronic^®^ F127, respectively, and aliquots (50 µL) were added to Eppendorf vials. In parallel, a set of aqueous solutions of Pluronic^®^ copolymers in PBS: water 50:50 *v*/*v* were also prepared in varying concentrations (2.5, 5, 25, 50, 250, and 500 mg/mL for Pluronic^®^ F68, 0.025, 0.05, 0.25, 0.5, 2.5, and 5 mg/mL for Pluronic^®^ F127; 5 mL each) and 4.95 mL of each solution was added to the Eppendorf vials containing the VES-GEM solution, to a final volume of 5 mL. The Pluronic^®^ copolymer micelles were also prepared in the same varying concentrations as controls, by dissolving the required amount of polymer in a PBS: water 50:50 *v*/*v* mixture. Next, the formulations were left overnight to equilibrate at room temperature in a hood and to allow ethanol evaporation, the latter in the case of the micelles was loaded with VES-GEM conjugates. Then, the surface tension of each solution (10 mL) was recorded using a Platinum ring in a tensiometer TD 1 Lauda (Fisher Scientific Hucoa, Madrid, Spain) [65].

### 2.8. Pluronic^®^/Soluplus^®^@VES-GEM Mixed Micelle Preparation

The effect of co-surfactant Soluplus^®^ was tested by preparing a series of Pluronic^®^/Soluplus^®^@VES-GEM mixed micelles with a varying Pluronic^®^ molar ratio and decreasing Pluronic^®^ F68 and Pluronic^®^ F127 concentration to compensate for the addition of the co-surfactant. Briefly, Pluronic F68^®^/Soluplus^®^@VES-GEM mixed micelles (0.375/0.375/1 and 0.75/0.375/1) and Pluronic F127^®^/Soluplus^®^@VES-GEM mixed micelles (0.375/0.375/1 and 0.75/0.375/1) were prepared as explained above, but with co-dissolving polymeric surfactants at the desired concentration in the ethanolic VES-GEM solution, and stirred for 3 h at RT before dropwise addition to PBS diluted in water (50:50 *v*/*v*, 15 mL) [66].

### 2.9. Micelle Characterization

#### 2.9.1. Micelle Size, ZP and PDI

The hydrodynamic size, polydispersity index (PDI) and zeta potential (ZP) of the different formulations (Pluronic^®^ F68/VES-GEM—molar ratio 0.75/1, 1.5/1, 3/1, 6/1; Pluronic^®^ 127/VES-GEM—molar ratio 0.75/1, 1.5/1, 3/1) were measured by dynamic light scattering (DLS) (Zetasizer Nano ZS, Malvern Instruments, Malvern, UK) [29]. Samples were not diluted and were previously analysed, both non-filtered and filtered, with a PTFE hydrophilic Scharlau syringe filter (25 mm, 0.4 µm). The experiments were run out in triplicates. The samples were equilibrated at 25 °C for 2 min and then analysed at 25 °C (at a scattering angle of 173°). The effect of centrifugation on the purification of Pluronic^®^ F68/VES-GEM (3/1) and Pluronic^®^ F127/VES-GEM (1.5/1) micelle formulations was assessed in a parallel set of experiments with formulations prepared in 1.5 mL Eppendorf^®^ tubes and either centrifuged at 4000 rpm, for 30 min, at 25 °C or at 12,000 rpm, for 20 min, at 25 °C (Eppendorf^®^ 5804R, Hamburg, Germany); then, the supernatant was carefully collected and the size, ZP, and PDI were measured through DLS (without filtration). The influence of sonication and ultrasonication on the size and PDI of the Pluronic^®^ F68/VES-GEM (3/1) and Pluronic^®^ F127/VES-GEM (1.5/1) micelles were first screened by submitting formulations to a 15 min sonication time, at RT, using an ultrasound J.P. Selecta bath (Laboquimia, Barcelona, Spain) and to different ultrasonication times and amplitudes, using a Branson Digital Sonifier with probe, Model 450, horn frequency 20 kHz, potency 400 W, 117 V, 50/60 Hz, intensity control 10–100% (Marshall Scientific, New Hampshire, NH, USA) [67]. For further optimization, the Pluronic^®^ F127 and Pluronic^®^ F127/VES-GEM (1.5/1) micelles were further tested at different sonication times (5 min, 10 min, 30 min, and 90 min), at RT, and distinct ultrasonication times (continuous mode for 3 s, 6 s, and 12 s, and for 24 s and 36 s in on/off mode (6 s on, 3 s off), at a 10% amplitude). To assess the influence of temperature on size, ZP, and PDI of the F68/VES-GEM (3/1) and Pluronic^®^ F127/VES-GEM (1.5/1) micelles, they were not diluted and were analysed at 4 °C and 37 °C, both non-filtered and filtered.

#### 2.9.2. Encapsulation Efficiency and Drug Loading

Encapsulation efficiency (EE, %) and drug loading (DL, %) were calculated after VES-GEM quantification through HPLC. Aliquots of the Pluronic^®^ F68/VES-GEM (3/1) and Pluronic^®^ F127/VES-GEM micelles were collected (100 µL) and diluted in ethanol (1:10 dilution), vortexed (Reax top model, Heidolph Instruments GmbH & Co. KG, Schwabach, Germany), and filtered before HPLC measurement using a PTFE hydrophilic Scharlau syringe filter (13 mm, 0.22 µm). The EE (%) of the VES-GEM conjugates in Pluronic^®^ micelles was calculated according to Equation (2). The DL (%) of the VES-GEM conjugates in Pluronic^®^ micelles was defined as the quotient between the amount of VES-GEM conjugate vs. the weighed amount of VES-GEM conjugate and Pluronic^®^ used to prepare the Pluronic micelles according to Equation (3) [68].
(2)$$EE\;(\%)=\frac{Quantified\;amount\;of\;VES-GEM\;conjugates\;in\;the\;micelles\;(mg)}{Weighted\;amount\;of\;VES-GEM\;conjugates\;in\;the\;micelles\;(mg)}$$
(3)$$DL\;(\%)=\frac{Quantified\;amount\;of\;VES-GEM\;conjugates\;in\;the\;micelles\;(mg)}{Weighted\;amount\;of\;VES-GEM\;conjugates+Pluronic®\;in\;the\;micelles\;(mg)}\;$$

For EE (%) calculation of the Pluronic^®^ F68/VES-GEM (3/1) and Pluronic^®^ F127/VES-GEM (1.5/1) micelles, both formulations were filtered (hydrophilic PTFE syringe filter (25 mm, 0.4 µm)) before aliquot collection. The calculation of the EE (%) of the Pluronic^®^ F68/VES-GEM (3/1) and Pluronic^®^ F127/VES-GEM (1.5/1) micelles after purification by centrifugation was carried out by centrifugation of the formulations (as described above), and the supernatant was collected without filtration, and following the steps explained previously.

#### 2.9.3. Transmission Electron Microscopy (TEM)

Non-filtered formulations either centrifuged or not were negatively stained with 2% phosphotungstic acid for 2 min and viewed on a TEM JEOL JEM1011 at 80 KV (JEOL, Peabody, MA, USA). Additionally, the morphology and structure of the formulations were analysed by FESEM (GeminiSEM, GEMINI 500, Zeiss, Oberkocken, Germany) 20 kV [31,68].

#### 2.9.4. UV-VIS Spectra

The spectra of the free VES-GEM conjugate in increasing concentrations (from 4 to 267 ppm) and of the Pluronic^®^ F68/VES-GEM (3/1), Pluronic^®^ F127/VES-GEM (1.5/1), Pluronic^®^ F68, Pluronic^®^ F127, Pluronic^®^ F68 + VES-GEM, and Pluronic^®^ F68 + VES-GEM were scanned (190–800 nm) using a UV–Vis spectrophotometer Agilent 8534, Waldbronn, Germany.

### 2.10. VES-GEM Conjugate Release Profile

The VES-GEM conjugate release profile was assessed through the dialysis method and the released amounts were quantified at predetermined time points through HPLC. Briefly, Pluronic^®^ F68/VES-GEM (3/1) and Pluronic^®^ F127/VES-GEM (1.5/1), at a VES-GEM concentration of 0.267 mg/mL (1 mL), were transferred into previously prepared dialysis membrane bags (MWCO 12.4 kDa; D9652-100FT, Sigma-Aldrich, St. Louis, MO, USA). The dialysis bags were sealed and placed inside cups containing release medium (PBS, pH 7.4, 0.48% *w*/*v* Tween 80, 15 mL) at two different pH values (5.0 and 7.4) [20,25] in the incubation shaker at 100 rpm/37 °C [36]. The release medium was sampled (0.25 mL) at predetermined time points (t = 0.5, 1, 2, 4, 8, 24, 48, and 72 h) for drug quantification and the volume was replaced with fresh medium. The collected aliquots were diluted to 50% with ethanol (the preferred solvent for VES-GEM solubilization) and filtered through a 0.22 μm membrane filter, and the VES-GEM content was quantified.

### 2.11. Solubilization Capacity of Tween 80

The capacity of Tween 80 to solubilize VES-GEM was evaluated using Tween 0.48% *w*/*w* in PBS (pH 7.4, 5 mL) solutions and adding an excess of VES-GEM conjugate (1 mg). The system was stirred for 2 h (300 rpm), followed by sonication (5 min) to enable the disruption of insoluble VES-GEM aggregates, and kept stirring for over 48 h (300 rpm), at RT. Aliquots were collected and filtered, and the VES-GEM content was quantified through HPLC. To ascertain if the drug release settings followed the sink conditions, the results were compared to the theoretical complete VES-GEM release from the formulations.

### 2.12. Stability of the Micelles

Stability testing of the different formulations prepared in physiological medium-mimicking conditions was conducted at both 4 °C and 37 °C in a PBS:water, 50:50 *v*/*v* medium. Briefly, Pluronic^®^/VES-GEM mixed micelles were incubated at 37 °C by placing them in the incubator shaker under gentle agitation, and at 4 °C by placing them in the refrigerator, for 4 weeks [20]. Samples were collected at predetermined time points, the VES-GEM content was monitored through HPLC, and physical parameters such as micelle size, ZP, and PDI were evaluated through DLS.

### 2.13. Blood Compatibility

Human blood (treated with citrate dextrose solution) from anonymized healthy donors (written informed consent) in accordance with Spanish legislation (Law 14/2007 on Biomedical Research [69]) was kindly provided by the Galician Transfusion Center (ADOS). Then, it was diluted with PBS [53] (final volume of 3.5% *v*/*v*). The Pluronic^®^ micelle formulations (0.1 mL) were placed in Eppendorf tubes containing diluted blood (0.9 mL), followed by incubation at 37 °C/1 h under gentle shaking (100 rpm) [70,71]. After the incubation period, the blood samples were centrifuged (2655 g/10 min) and the supernatants transferred in duplicate to a 96-well plate to measure the absorbance of the released haemoglobin by lysed erythrocytes, in a plate reader (540 nm) (FLUOstar optima, BMG LabTech, Ortenberg, Germany). Triton X-100 1% *v*/*v* was selected as a positive control (hemolytic activity) and DPBS and ethanol 2.5% *v*/*v* solutions were used for the negative controls. The experiment was run in triplicate. Hemolytic activity was defined regarding the extent of haemoglobin release from lysed erythrocytes and calculated in percentage (%) according to Equation (4):(4)$$Hemolysis\;(\%)=\frac{A_{S\;}-A_{N}}{A_{P\;}-A_{N}}\;\;$$

A_S_ stands for sample absorbance, A_N_ is the negative control absorbance, and A_P_ represents positive control absorbance.

### 2.14. Cell Viability

Human pancreatic cancer BxPC3 cells (ATCC CRL-1687™, ATCC, USA) were cultured in RPMI-1640 medium supplemented with 10% of FBS and 1% antibiotics (10,000 U/mL penicillin and 10,000 µg/mL streptomycin) (Sigma Aldrich, St. Louis, MO, USA) in an atmosphere of 5% CO_2_ and 95% RH at 37 °C. The cytotoxicity of the drugs/conjugate, blank micelles, and drug-loaded micelles was determined using a Quant-iT PicoGreen dsDNA assay kit (ThermoFisher, Waltham, MA, USA). BxPC-3 cells (80% confluency) were expanded in RPMI-1640 supplemented with 10% of foetal bovine serum and 1% of penicillin/streptomycin, and seeded at a density of 2 × 10^4^ cells/well into a 96-well plate and incubated with culture medium for 24 h at 37 °C in a 5% CO_2_, 95% RH humidified incubator. Then, the solutions of GEM, Pluronic^®^ F68 micelles, Pluronic^®^ F127 micelles, Pluronic^®^ F68/VES-GEM (3/1) micelles, and Pluronic^®^ F127/VES-GEM (1.5/1) micelles in PBS diluted in water (50:50 *v*/*v*), and VES-GEM (in DMSO), were prepared, filtered (Biofil sterilized syringe filter, 30 mm, 0.22 μm PES membrane; Barcelona, Spain), and added to the seeded cells; carried out in quadruplicate. Different concentrations were tested, namely 100 µM, 50 µM, 25 µM, 10 µM, and 1 µM of VES-GEM (or equivalent; this means a corresponding blank formulation to the same dilution as its VES-GEM-loaded counterpart). Culture medium was used as a control. The cells were seeded and the cell plate was incubated for 24 h under the same conditions and the cells were monitored under a Nikon Eclipse TS100 microscope, equipped with a DXM 1200 digital camera (Nikon, Tokyo, Japan). Cell proliferation was analysed in accordance with the PicoGreen protocol. In brief, the culture medium was withdrawn from the wells and the cells were washed twice with PBS. Then, 200 μL of DNAase-free water was added to each well and the plates were then subjected to three freeze–thaw cycles. Then, 100 μL of the DNA samples were incubated with 100 μL of working solution and allowed to react for 3 min while protected from light. Finally, the DNA content was read in a fluorescence microplate reader (FLUOstar OPTIMA microplate reader, BMG Labtech, Germany) (λexc 485 nm; λem 530 nm). A DNA standard curve was used to quantify the amount of DNA in each sample. Cell viability was calculated using Equation (5), as follows.
(5)$$Cell\;viability\;(\%)=\frac{F_{exp\;}-F_{blank}}{F_{control\;}-F_{blank}}\;$$

F*_exp_* stands for sample fluorescence intensity, F*_blank_* represents the blank, and F*_control_* stands for control fluorescence intensity (non-treated cells). For the IC_50_ calculation, interpolation with the plotted line obtained by GraphPad^®^ Prism^®^ software (version 9.5; San Diego, CA, USA) was carried out and the y_intercept_, slope and R squared parameters were obtained.

### 2.15. In Vitro Cell Uptake

Pluronic^®^ F68 and F127 micelles were prepared as described in Section 2.6. Fluorescently labelled micelles were prepared as follows. Nile red (ThermoFisher GmBH, Kandel, Germany) was dissolved in DMSO (0.4 mg/mL) and stirred for 5 h. Then, 100 µL of the Nile red solutions were added to 1.15 mL of ethanol, vortexed, and added dropwise to 3.75 mL micelle dispersions under stirring and left overnight in a hood, protected from light for the ethanol to evaporate (final Nile red concentration: 1.07 µg/mL, DMSO < 3% *v*/*v*). BxPC3 cells were cultured following the method described above, seeded on 8-well glass slides (Lab-Tek II chamber slides; Thermo Scientific, Waltham, MA, USA) at a density of 1 × 10^5^ cells/well and incubated overnight. Then, the cells were exposed to Nile red-loaded micelles (0.1 mg mL^−1^) for 2 h. The cells were then washed thrice with PBS, fixed with a 4% paraformaldehyde (100 µL/well) for 10 min, and rinsed three more times with PBS. Then, the cells were incubated with Triton X-100 (0.2% in PBS) for 5 min, followed by rinsing thrice with PBS. Subsequently, the samples were mounted using DAPI ProLong gold (Molecular Probes; Eugene, OR, USA), covered with a glass coverslip, and kept at −20 °C until observation. Confocal images were captured using a Leica confocal TCS-SP5 microscope (Leica Microsystems; Wetzlar, Germany).

### 2.16. Statistical Analysis

Statistical data analysis was carried out using GraphPad^®^ Prism^®^ software version 9.5 (San Diego, CA, USA). All results were expressed as mean ± standard deviation and were analysed using ANOVA followed by a post hoc Tukey honest significance test, where appropriate *p* values < 0.05 were considered statistically significant.

## 3. Results and Discussion

### 3.1. Synthesis and Characterization of VES-GEM

VES was chosen for bridging with antimetabolite GEM. VES is a vitamin E derivative with antioxidant and anticancer properties bearing a terminal carboxyl group that can be exploited for conjugation with GEM. The VES-GEM conjugate, which may perform as a dimeric prodrug, was synthesized by amidation of the carboxylic acid group of VES with the primary amine group of GEM, resulting in a stable and cathepsin B-responsive amide bond. The final yield (ca. 53%) was in accordance with previously described results [34]. This reaction process was adopted by virtue of its relatively simple procedure and considerable yield. Predicted ^13^C NMR (Appendix A) and ^1^H NMR (Appendix A) spectra were obtained using the Chemicalize and MarvinView software models. The infrared spectrum of the VES-GEM conjugate showed absorbance at 3500–3150 (3297 peak), 2927, 2854, 1727, 1658, 1488, and 1241 (Appendix A). The presence of typical peaks at 1727, 1488, and 1241 cm^−1^ and the absence of peaks ascribed to amine groups within the 3500–3150 cm^−1^ range revealed that an amide form was formed through the reaction of the carboxyl group of VES with the amine group instead of the hydroxyl group of GEM. The ^1^H NMR spectrum of VES-GEM is shown in Appendix A. ^1^H NMR (300 MHz, CHCl_3_-d_6_) signals include: 9.83 (s, 1 H, H4), 7.86 (d, 1 H, H6), 7.46 (d, 1 H, H5), 5.97 (s, 1 H, H3), 5.32 (s, 1 H, H1), 4.93 (s, 1 H, H5″), 4.08–3.98 (m, 3 H, H5′ and H4′), 3.91 (d, 1 H, H3′), 2.90 (s, 2 H), 2.59 (s, 2 H), 2.09 (s, 4 H, H7 and H8), 2.00 (d, 8 H), 1.66 (s, 3 H), 1.56 (m, 4 H), 1.40 (s, 3 H), 1.27 (m, 13 H), 1.16 (s, 4 H), 1.09 (s, 4 H), and 0.88 (d, 16 H). The presence of a peak near 10 ppm (singlet), ascribed to the amide group of VES-GEM, and the absence of a peak near 7 ppm referring to the H signals of the free amine group (N^4^ of the pyrimidine ring of GEM) attested to the formation of an amide bond between VES and GEM and the formation of a VES-GEM conjugate. The ^19^F NMR spectrum showed a typical shift ~ −120 ppm for F_2_CH-R, in which R comprises two or more methyl groups (Appendix A). The MS spectrum of VES-GEM showed a main peak near *m*/*z* (%): 430 [M+] (Appendix A). The presence of the molecular ion (peak with strongest intensity) near *m*/*z* (%): 430 represents free vitamin E released from the conjugate, and the remaining GEM linked to succinate moiety appears near *m*/*z* (%): 413. Altogether, the results confirmed the successful synthesis of the VES-GEM prodrug conjugate. The stability assay regarding repetition of the ^1^H NMR spectrum of the stored VES-GEM showed no significant alterations in terms of peak distribution, ensuring its chemical stability during long-term storage at 4 °C (Appendix A). Using ChemDraw, the conjugate was characterized regarding pKa (~11.72), LogS (−5.25), and LogP (5.08), which can indicate a basic behaviour, low water solubility, and lipophilic nature. A more extensive chemical characterization was obtained by referring to the Chemicalize and MarvinView software models (Table 2).

Accordingly, the molecular weight was calculated to be ~776 g/mol, with isoelectric point of 6.26, which shows that VES-GEM is electrically neutral at pH 6.26, close to physiological pH. A log_10_ of the partition coefficient (P) (logP) is >8.5, hence it is hydrophobic and poorly soluble in aqueous medium (<0.01 mg/mL), showing higher affinity for the organic solvent phase (Appendix A). At pH 7.4, the VES-GEM conjugate has an overall neutral charge and higher prevalence of H bond acceptor sites. The predicted ^1^H NMR spectrum was concordant to the one obtained experimentally. Hydrophilic and hydrophobic regions, logP and logD, are shown in Appendix A and the predicted 3D structure of the conjugate is shown in Appendix A.

### 3.2. Stability of the Free Drug and Conjugate

The chemical stability of both GEM and VES-GEM in solution was assessed via HPLC over several days and at both 4 and 37 °C. Stability testing of GEM showed no appreciable differences between day 0 and day 55, with suggests GEM is stable at both temperatures in a period of almost two months (Figure 3A). GEM was also considered as photostable, as shown by the minimal variation in concentration (Figure 3B). On the other hand, the VES-GEM conjugate underwent significant degradation upon incubation at 37 °C as opposed to minimal variation in the concentration of the conjugate at 4 °C (Figure 3C). Photostability studies suggested that the VES-GEM conjugate is not photodegradable (Figure 3D). The stability at RT of VES-GEM showed a substantially minor decrease in VES-GEM content after 8 days when compared to 37 °C (Appendix A).

Following cellular uptake, the micelles are expected to enter the endolysosomal compartment which is characterized by an acidic microenvironment (pH~5). An attempt was made to test the stability of free VES-GEM at pH = 5; the VES-GEM concentration at 37 °C decreased significantly after one week of incubation while the concentration remained almost constant when tested at 4 °C. These findings pointed to temperature as the key variable affecting the stability of the VES-GEM conjugate (Appendix A). In the case of GEM, since the drug is activated partially inside the endolysosomal compartment, either through low pH action or enzymatic activity, it would be of interest to test the stability of GEM at the expected pH value of these surroundings. The results showed no appreciable concentration variation after the incubation period, which is suggestive of the stability of GEM at pH levels typical of the endolysosomal compartment (Appendix A).

### 3.3. Pluronic^®^/VES-GEM Conjugate Micelle Preparation and Characterization

VES-GEM is overall an amphiphilic molecule in which the VES aliphatic tail contained in the lipidic portion is hydrophobic, whilst the GEM portion is more polar and contains -OH, -F, and -NH_2_ groups [72]. The lipophilic moiety of the VES-GEM conjugate prodrug system may allow successful encapsulation in the hydrophobic core of the Pluronic^®^ micelles by conferring hydrophobicity and also additional antioxidant and anticancer properties.

First, a set of Pluronic^®^/VES-GEM micelles with different polymer-to-conjugate molar ratios was prepared either with Pluronic^®^ F68 (0.75/1, 1.5/1, 3/1, and 6/1), comprehended in a 0.2–1.8% *w*/*v* range for Pluronic^®^ F68, and Pluronic^®^ F127 (0.75/1, 1.5/1, 3/1), and 0.3–1.3% *w*/*v* for Pluronic^®^ F127 (Table 3 and Figure 4A,B). The concentration of VES-GEM was fixed to 0.267 mg/mL and the copolymer concentration was varied. PBS:water 50:50 *v*/*v* was chosen as the formulation medium as PBS may favour micellization when compared to water [73] and water was used to decrease the osmolarity of the solution as the micelle system may already contribute to increased osmotic pressure. TEM images of the blank and VES-GEM loaded micelles are compiled in Appendix A.

VES-GEM loading was accompanied by an increase in the hydrodynamic diameter of Pluronic^®^/VES-GEM micelles, in agreement with reports on other hydrophobic drugs such as ibuprofen, aspirin, and erythromycin [74]. Pluronic^®^ F68/VES-GEM micelles showed sizes between 150–200 nm and ZP was slightly negative for the formulations with polymer ratios 1.5, 3 and 6 (Figure 4C–E). The lowest PDI values were shown for the 1.5/1 and 3/1 Pluronic^®^ F68/VES-GEM micelles, as increasing the copolymer concentration may increase the solubility of the VES-GEM prodrug conjugate and contribute to a more homogeneously dispersed population, and optimal PDI values are reported for 6/1 Pluronic^®^ F68/VES-GEM micelles (<0.2).

Regarding the Pluronic^®^ F127/VES-GEM micelles, varying the molar ratio resulted in micelles in the range of 100–200 nm with a slightly negative surface charge, and the same was verified for Pluronic^®^ F68/VES-GEM micelles (Figure 4F–H) [52]. PDI values for Pluronic^®^ F127/VES-GEM micelles were >0.8 for all formulations indicating the heterogeneity of the population. To select the formulations for the next step of characterizations, a strategy was devised by (1) trying to use the lowest possible amount of copolymer whilst (2) maintaining appreciable VES-GEM solubility capacity and stability, and (3) the least polydisperse formulation. Pluronic^®^ F68/VES-GEM with molar ratios of 3/1 showed decreased agglomeration and improved stability and acceptable PDI. Pluronic^®^ F127/VES-GEM with molar ratios of 1.5/1 showed an absence of agglomeration when compared to the 0.75/1 counterpart, and evidenced an increased ability to solubilize VES-GEM for equal surfactant concentration when compared to Pluronic^®^ F127, which may be due to the increased length of the PPO segment. Pluronic^®^ F68/VES-GEM 3/1 and Pluronic^®^ F127/VES-GEM 1.5/1 micelles were therefore chosen, and the TEM images showed spherical structures corroborating the formation of nanosized micelles (Figure 4I,J, Figure 5A–H and Appendix A). Both selected formulations displayed high encapsulation efficiency (EE), namely 96.33 ± 7.80% for Pluronic^®^ F68/VES-GEM (3/1) and 96.48 ± 4.82% for Pluronic^®^ F127/VES-GEM (1.5/1), and low drug loading, 2.90 ± 0.04% and 3.83 ± 0.02% for Pluronic^®^ F68/VES-GEM (3/1) and Pluronic^®^ F127/VES-GEM (1.5/1) micelles, respectively. This suggests that the micelles are able to solubilize the majority of the VES-GEM conjugate while maintaining a relatively low drug-to-polymer ratio. Nevertheless, some polydispersed aggregates were visually detected, mainly in Pluronic^®^ F68/VES-GEM micelle formulation. This observation warrants a purification procedure that refers to the removal of not only the free non-encapsulated drug, but also any free polymer aggregations as well as drug-loaded micelle aggregation [75]. Hence, the VES-GEM content in both the Pluronic^®^ and VES-GEM micelles was quantified through HPLC after filtration and compared to non-filtered formulations, as adjusted for EE (%). Filtration removed all visible aggregates of Pluronic^®^ F68/VES-GEM (3/1) micelles and the results showed a considerable decrease in the VES-GEM concentration to half in both formulations. This may be explained by the removal of non-encapsulated VES-GEM present in the aggregates. The Pluronic^®^ F127/VES-GEM (1.5/1) micelles still showed slightly higher EE (%), which may be explained by the lower CMC value, larger monomer chain length and molecular weight, higher size, and the increased number of -PPO- units which improve hydrophobic VES-GEM solubilization and lead to stronger encapsulation (Appendix A) [76,77].

A complementary study was performed by increasing substantially the concentration of either Pluronic^®^ F68 (4.3% *w*/*v*) or Pluronic^®^ F127 (3.25% *w*/*v*) to assess if structures with lower size and PDI values could be obtained (Appendix A and Appendix A). However, the structures formed had size ranges in micrometres and higher PDI and were not considered in further studies.

The ability of VES-GEM to self-assemble in nanostructures was also explored following the same method and inspired by previous studies which reported nanostructures made of lipid–GEM conjugates without additional surfactant addition [14,27]. The formulation showed large aggregates and a particle size of 286 ± 40.26 nm, ZP of −0.4 ± 0.75 mV, PDI = 0.36 ± 0.14, and a yield below 10% due to its low water solubility and was not further explored.

After this first set of experiments, the selected formulations—Pluronic^®^ F68/VES-GEM (3/1) with 0.86% *w*/*v* of Pluronic^®^ F68, and Pluronic^®^ F127/VES-GEM (1.5/1) with 0.65% *w*/*v* of Pluronic^®^ F127—were prepared again and the influence of filtration was assessed and further compared to the blank formulations. The stable Pluronic^®^/VES-GEM conjugate micelles could be obtained, although the Pluronic^®^ F127/VES-GEM conjugate micelles showed the best colloidal stability without any visible aggregates, as reported before. DLS measurements revealed that the average hydrodynamic size of Pluronic^®^ F68/VES-GEM micelles was 140.01 ± 0.28 nm, four-fold higher when compared to the blank Pluronic^®^ F68 micelles (34.24 ± 13.11 nm) (Table 3).

The same tendency was observed for Pluronic^®^ F127/VES-GEM micelles (136.67 ± 25.70 nm vs. 25.57 ± 0.47 nm). Filtration of the micelles resulted in a decrease in size more pronouncedly for the Pluronic^®^ F127/VES-GEM micelles. A decrease in PDI values was only observed for the Pluronic^®^ F68/VES-GEM micelles. The incorporation of the VES-GEM conjugate substantially increased the size of the micelles for both cases. The intensity mode (%) of the blank Pluronic^®^ F68 micelles showed a polydisperse population with a predominant population near 10 nm, which shifted to values closer to ~100 nm when the micelles were loaded with VES-GEM (Figure 6A,C). A more complex distribution was noted for the Pluronic^®^ F127/VES-GEM micelles, which show three distinct populations and a maximum intensity peak shift to values near 100 nm for their blank counterparts (Figure 6A,C). By changing the DLS measurement mode to number (%), all the formulations appeared to be monodispersed, with the blank micelles nearing 10 nm (max. intensity peak). The sizes of the Pluronic^®^ F68/VES-GEM micelles and the Pluronic^®^ F127/VES-GEM micelles were in the range of 10–80 nm and 50–300 nm, which revealed the most frequent subpopulation and attenuates the weight of the rare but larger particles in intensity mode analysis (Figure 6B,D). The VES-GEM-loaded micelles have a slightly negative surface charge (−5.38 ± 1.60 mV for Pluronic^®^ F68/VES-GEM and −1.54 ± 1.08 mV for the Pluronic^®^ F127/VES-GEM conjugate micelles), which is in accordance with values reported in the literature [53,54]. The absorbance spectra for the VES-GEM conjugate in ethanol at different concentrations and the GEM, VES-GEM, and micelle formulations are shown in Figure 6E,F, and the appearance of the non-filtered Pluronic^®^ F68/VES-GEM and Pluronic^®^ F127/VES-GEM micelles is depicted in Figure 6G.

It is also well known that temperature is able to influence parameters such as the size of micelles. A set of Pluronic^®^/VES-GEM formulations were reanalysed by measuring size and ZP at 4 and 37 °C. Hydrodynamic size generally increased at 37 °C for low polymer concentrations. The increase in size was also accompanied by an increase in PDI for most of the formulations, and the surface charge increased to almost neutral values when the temperature of the measurement was set at 37 °C (Figure 7). However, for higher polymer-to-VES-GEM molar ratios, a slight decrease in the hydrodynamic diameter was observed for both formulations. This behaviour has previously been explained by a reduction in CMC value and increased micellization capabilities as temperature rises, on account of the dehydration of the -PEO- blocks and the increased hydrophobicity of the chains [76]. Additionally, increasing the temperature may lead to the dehydration of the -PPO- blocks, which increases core hydrophobicity, which may enable the formation of more compact micelles evidencing smaller sizes [74]. The extent of hydrophobic interactions in the micelle core may be responsible for decreasing CMC together with improving the stability of the system [78]. According to reports in the literature, the increase in temperature may also contribute to hydrophobic drug solubilization in micelles not only because of CMC reduction but also related to micellar growth [79].

Although filtration of agglomerates of the formulations may help purify the nanosystem, it can also retain some micelles and the overall VES-GEM encapsulation may not be entirely reliable. In addition to filtration as a purification technique, centrifugation has been widely described in the literature as an efficient method to remove unloaded drug and obtain purified nanosystems [75]. While centrifugation may also have some disadvantages, as the applied strong forces may impact the structure and content of nonrigid dynamic colloidal systems, such as micelles, triggering early drug leakage or caking, it is still considered as a suitable method for the purification of micelles [80]. The impact of the centrifugation technique on the size, ZP, PDI, EE (%), and morphology of the Pluronic^®^/VES-GEM micelles was therefore assessed by two centrifugation settings: 4000 rpm/30 min and 12,000 rpm/20 min, both at 25 °C, and the results were compared with the original nonfiltered and filtered counterparts (both non-centrifuged). The original Pluronic^®^/VES-GEM micelles were prepared on different days for each centrifugation procedure. The size of the centrifuged formulations was, in most cases, in between those of the non-filtered and filtered ones, which indicates the removal of aggregates representing populations with larger sizes (Figure 8A,D). Both centrifugation settings exhibited the same pattern in terms of size and PDI oscillation, but the filtration yields a less polydisperse population when compared to centrifugation (Figure 8B,E). ZP values for both formulations were slightly negative without major changes after both techniques (Figure 8C,F). Regarding size distribution analysis, filtration and centrifugation decreased the intensity peak of populations > 1000 nm, which was more pronounced in the filtration group and for both the Pluronic^®^ F68/VES-GEM (3/1) and Pluronic^®^ F127/VES-GEM (1.5/1) conjugate micelles. Overall, both purification techniques tested can eliminate larger populations and concentrate the size distribution within the 10–1000 nm range (Figure 8G–L).

The EE (%) values of the Pluronic^®^ F68/VES-GEM and Pluronic^®^ F127/VES-GEM micelles centrifuged at 4000 rpm were 29.90 ± 7.94% and 24.07 ± 2.80%, respectively; meanwhile, after 12,000 rpm, the centrifugation the values decreased to 10.49 ± 2.08 and 7.58 ± 0.14%, respectively (Figure 9A,F). The physical aspects of the Pluronic^®^ F68/VES-GEM and Pluronic^®^ F127/VES-GEM micelles after 4000 rpm/30 min/T = 25 °C centrifugation are shown in Figure 9B–E and Figure 9G–J, respectively. These values are lower than for non-centrifuged micelles, possibly due to the elimination of free VES-GEM and large aggregates, and, at 12,000 rpm, the micelles may even collapse and disentangle.

As the PDI value of the Pluronic^®^ F127/VES-GEM micelles (1.5/1) was ca. 1, a strategy was devised to test the influence of sonication and ultrasonication on the PDI and size of the micelles [67,81]. First, initial screening for both the Pluronic^®^ F68/VES-GEM (3/1) and Pluronic^®^ F127/VES-GEM (1.5/1) micelles was undertaken to compare non-sonicated and non-ultrasonicated formulations with their sonicated and ultrasonicate counterparts, regarding size and PDI, as well as to establish the optimal range conditions of ultrasonication time and the amplitude of the ultrasonication procedure. Amplitude values > 10% would cause excessive bubbling and foaming and were not pursued. A milder amplitude value was selected (10%) for ultrasonication, as the system showed increased stability and, for ultrasonication times of 24 s and 36 s, an on/off pulse mode was chosen to prevent the overheating of the system. As shown in Appendix A, both sonication for 15 min and ultrasonication for 6 s increased the size and PDI of the Pluronic^®^ F68/VES-GEM micelles, but yielded a better distribution profile for Pluronic^®^ F127/VES-GEM micelles. Hence, modulating the PDI of the Pluronic^®^ F127/VES-GEM micelles through sonication and ultrasonication was further assessed by varying sonication (Table 4) and ultrasonication time (Table 5) in order to determine the optimal conditions that underscore the lowest polydispersity of the Pluronic^®^ F127/VES-GEM micelle population.

According to the results, the PDI of the Pluronic^®^ F127/VES-GEM micelles showed a tendency to decrease with increases in sonication time, and more remarkably so in the case of ultrasonication (0.717 ± 0.012 vs. 0.294 ± 0.013, filtered and after 90 min of sonication and 36 s of ultrasonication). Interestingly, at the same time points, the size of the Pluronic^®^ F127/VES-GEM micelles was fairly similar to the non-sonicated (66.45 ± 3.66 nm vs. 131.33 ± 23.06 nm, filtered, respectively) and non-ultrasonicated ones (186.50 ± 4.78 nm vs. 131.33 ± 23.06 nm, filtered, respectively), whilst the difference was more pronounced for the non-filtered Pluronic^®^ F127/VES-GEM micelles subjected to ultrasonication, in which non-filtered Pluronic^®^ F127/VES-GEM micelles showed a size of 395.83 ± 17.97 nm after 36 s of ultrasonication as opposed to 2146.33 ± 276.06 nm for the group not exposed to ultrasonication. In addition, just 3 s of ultrasonication can lead to a two-fold decrease in the PDI of Pluronic^®^ F127/VES-GEM micelles. These results may be due to the disruption of larger structures and aggregates to form a smaller, more narrowly distributed micelle population, and this is expectedly shown to be more efficient in the ultrasonication group as ultrasonication is a more potent sonication technique [67,81]. The maximum duration of the ultrasonication experiment was kept at 36 s, not only to avoid reagglomeration of the system and potential instability due to temperature rise, but also due to the potency of the ultrasonication method. Furthermore, maximum PDI reduction was achieved for the intermediate time point of sonication (10 min), whilst in the case of ultrasonication, a decrease in PDI for the Pluronic^®^ F127/VES-GEM micelles was reported when comparing the non-ultrasonicated, 3 s sonicated, and 36 s sonicated micelles, suggesting, in the context of the time points selected for the experiment, that progressively lower PDI values may be obtained by increasing the ultrasonication time. Conversely, the non-loaded ultrasonicated Pluronic^®^ F127 micelles showed an increasing PDI as the duration of ultrasonication increased, which was not evident in the sonication experiment. Analysis of size through intensity (%) shows deviation towards the lower size range for the filtered ultrasonicated (36 s, 6 s on 3 s off) Pluronic^®^ F127/VES-GEM micelles vs. the non-filtered ones (Appendix A), and similar to the non-ultrasonicated ones (Figure 6A,C); however, with a less intense peak for the non-filtered group, showing the effect of ultrasonication in terms of decreasing particle size, probably through deagglomeration. The number (%) mode of the Pluronic^®^ F127/VES-GEM micelles subjected to ultrasonication showed peak deviation towards the ~100 nm range for the non-filtered group (Appendix A), as opposed to the non-ultrasonicated micelles (Figure 6B,D). Overall, ultrasonication seems to decrease the PDI of Pluronic^®^ F127/VES-GEM micelles while not inducing substantial variation in particle size, and could be of interest in the future as a simple and rapid strategy for micelle optimization by decreasing PDI. However, the structures obtained warrant further characterization regarding the morphology, structural integrity, and chemical stability of the VES-GEM prodrug, and the non-sonicated and non-ultrasonicated Pluronic^®^ F127/VES-GEM micelles were still preferred for the following experiments.

### 3.4. Critical Micelle Concentration of the Pluronic^®^/VES-GEM Micelles

In the case of the Pluronic^®^/VES-GEM conjugate micelles, surface tension increased when compared to the blank micelles because these micelles had larger sizes and less surfactant was available at the air–water interface, rendering more surfactant available for micelle formation (Appendix A). The broader difference in surface tension values was observed in the Pluronic F68^®^ micelle setting. The observed CMC was identified in the 0.1–2.5 mM range and, after 10 mM, micellization was favoured, which was in accordance with the literature (0.48 mM) [82]. The difference in surface tension between the blank and Pluronic^®^ F127/VES-GEM conjugate micelles was less substantial, as Pluronic^®^ F127 has a very strong capability to reduce surface tension and very low CMC, while the values in the literature may vary (0.0028 mM to 0.45 mM) [63,83]. The surface tension/polymer concentration plotting made it possible to identify the CMC for the blank and VES-GEM-loaded Pluronic^®^ F127 micelles close to 0.15 mM, which falls within the reported range. Thus, in the Pluronic^®^ F127 range of concentrations tested to prepare the Pluronic^®^ F127/VES-GEM micelles, the copolymer was above the CMC.

### 3.5. Addition of Co-Surfactant

Addition of a co-surfactant to micelle systems may help improve overall stability and drug solubility. Recently, Pluronic^®^ F68/Soluplus^®^ mixed micelles carrying pterostilbene were prepared as a strategy to leverage Pluronic^®^ F68 drug loading and stability [84]. Inspired by this, a series of Pluronic^®^/Soluplus^®^@VES-GEM mixed micelles were prepared, keeping the Soluplus^®^ concentration fixed and increasing the concentration of the Pluronic^®^ constituent, especially aiming at improving Pluronic^®^ F68 stability. Soluplus^®^ is an amphiphilic graft copolymer—polyvinyl caprolactam-polyvinyl acetate-polyethylene glycol (PEG)—which shows the capacity to self-assemble into micelles under extremely low CMC (0.1 μM) [85]. In addition to its self-assembly features, it also shows multidrug resistance (MDR) reversal activity by inhibiting efflux pump P-gp. In line with this, we attempted to decrease the Pluronic^®^ concentration and added Soluplus^®^ at a fixed concentration (14.72 mg/mL, 1.47% *w*/*v*), preparing four different formulations: Pluronic F68^®^/Soluplus^®^@VES-GEM (0.375/0.375/1 and 0.75/0.375/1) and Pluronic F127^®^/Soluplus^®^@VES-GEM (0.375/0.375/1 and 0.75/0.375/1) mixed micelles (Appendix A). The Pluronic F68^®^/Soluplus^®^@VES-GEM mixed micelles showed improvements in stability as fewer agglomerates were detected, when compared to the other two formulations, which may be due to the enhanced solubilizing properties of Soluplus^®^. All formulations evidenced a size of 80–120 nm (Figure 10A) and very low PDI, as opposed to the single Pluronic^®^/VES-GEM micelles, which reflected the propensity of Soluplus^®^ to create monodispersed micelle populations (Figure 10B) and a close-to neutral surface charge (Figure 10C). The 0.75/0.375/1 mixed micelle had a decreased PDI and an increased VES-GEM encapsulation (Figure 10D), as the total surfactant-to-conjugate molar ratio was augmented.

### 3.6. Drug Release from Pluronic^®^/VES-GEM Micelles

Since the theoretical aqueous solubility of VES-GEM is very low (<0.01 mg/mL), the solubility of the conjugate in the presence of Tween 80 was investigated in order to infer whether the drug release experiment was in accordance with the sink conditions. The maximum VES-GEM concentration released from the formulation was expected to be 17.0–17.2 ppm. The measured solubility of VES-GEM in Tween 0.48% *w*/*w* in PBS (pH 7.4, 5 mL) was 57.8 ± 0.1 ppm, which was four-fold higher than the concentration that would correspond to a 100% release from the Pluronic^®^ F68/VES-GEM and Pluronic^®^ F127/VES-GEM micelles. These results show that, while ideal sink conditions are not fully accomplished, the medium was still able to fully solubilize VES-GEM if a 100% release occurred. DLS measurements were carried out to ascertain if Tween 80/VES-GEM micelles could be formed in the medium, and the results indicated that structures with a size of ~15 nm were formed in the blank Tween 80. When VES-GEM was added as a white powder directly to the surfactant dispersions, the size was maintained, while the PDI decreased to values below 0.2 (Appendix A), which suggested VES-GEM loading, as further corroborated by ethanol dilution of aliquots of the medium in the drug release experiment described below.

The drug release assay was conducted for 1 week. After 72 h, the VES-GEM cumulative release from the Pluronic^®^ F127/VES-GEM (1.5/1) micelles was 54.02 ± 1.08% at pH = 7.4 and 35.02 ± 19.53% at pH = 5, as opposed to minimal cumulative release from the Pluronic^®^ F68/VES-GEM (3/1) micelles (<4% at both pH) (Figure 11). Interestingly, in the first 24 h, the release of VES-GEM from Pluronic^®^ F127/VES-GEM was more prominent at pH = 5, which may indicate that the system may display mild pH-responsiveness features underscoring a more accelerated VES-GEM release profile in physiological acidic environments, considering the first 24 h of the assay. Whilst the Pluronic^®^ F127/VES-GEM micelles were able to show a remarkably better release profile than Pluronic^®^ F68/VES-GEM, it represented only half of the loaded VES-GEM. In both formulations, these results can be attributed in part to the high affinity of the hydrophobic vitamin E tail to the hydrophobic -PPO- segments of Pluronic^®^ F68 and Pluronic^®^ F127, thereby increasing VES-GEM solubility and retention inside the micelle core and attenuating its release. It is possible that hydrophobic VES-GEM prodrugs can themselves establish hydrophobic interactions and other non-covalent bonds (such as hydrogen bonds) which further strengthen the stability of the system and prevent VES-GEM leakage [23]. Another major factor consists of the high lipophilicity of VES-GEM which restrains release to the aqueous release medium. For this reason, Tween 80 was used as surfactant to improve VES-GEM solubility and enable near-sink conditions. Other authors have also tried the addition of an organic solvent, such as methanol or ethanol, to the aqueous medium, which may help in part to increase drug release, but these solvents are not representative of the in vivo milieu [37]. Furthermore, the slow release profile of the formulations (>7 days) suggested they constitute stable and suitable systems for VES-GEM prodrug delivery to pathological sites. Other similar studies have reported only moderate GEM prodrug release, in the case of DSPE-PEG/TPGS mixed micelles loaded with C_18_-GEM and tested in dialysis cassette (MWCO 20 kDa) against PBS medium, half of the initial prodrug content is released after 12 h [20]. C_18_-GEM-loaded PEG/PLA micelles were tested using dialysis bags against PBS with Tween 80 0.5% showing <30% release of C_18_-GEM after 72 h [23].

This drug release assay protocol was adopted after a preliminary test in which it was observed that ethanol dilution (1:1) of the aliquots collected from the medium showed higher VES-GEM content when compared to non-diluted and water-diluted (1:1) aliquots, which may suggest that ethanol may help in disrupting Tween 80/VES-GEM structures and improve the accessibility of VES-GEM to be quantified through HPLC, as well as avoiding VES-GEM precipitation. Tween 80 was chosen as the most representative surfactant used in the literature in this setting [20,23,25].

### 3.7. Stability of Pluronic^®^/VES-GEM Micelles

The stability of the Pluronic^®^/VES-GEM micelles was evaluated both physically by monitoring size, PDI, and ZP, and chemically, by pH measurement of the formulations and by quantifying VES-GEM content through HPLC at 4 and 37 °C. After 28 days, both formulations showed remarkably larger sizes at 37 °C when compared to 4 °C (Figure 12A), while an opposite tendency was verified for PDI, which was inferior at 37 °C (Figure 12B). This change may be explained by possible increased aggregation phenomena and instability at higher temperatures for longer time periods, which could decrease the heterogeneity of the population and lead to a narrower distribution and a less polydisperse population. ZP values were slightly negative throughout the duration of the experiment (Figure 12C).

Regarding chemical stability, the pH of the formulations suffered minimal variations, and the initial pH was similar for both micelles (~7.6) (Appendix A). However, the Pluronic^®^/VES-GEM micelles showed limited chemical stability, especially at 37 °C and after the first 2 days. These results are in accordance with the decreased stability of the free VES-GEM conjugate at 37 °C, shown before. Interestingly, the VES-GEM concentration detected increased two-fold after 28 days in the Pluronic^®^ F68/VES-GEM (3/1) micelles at 37 °C, which may be explained by the formation of visible aggregates and precipitates that compromise the viability of HPLC quantification by unwanted quantification of the concentrated aggregates of the VES-GEM conjugate (Figure 12D). On the other hand, the Pluronic^®^ F127/VES-GEM (1.5/1) micelles evidenced an increased ability to protect the VES-GEM conjugate and a more homogeneous appearance without visible aggregates (Figure 12D and Appendix A). After that, the Pluronic^®^ F127/VES-GEM micelles were kept at RT for 2 months, and after this period, their size, PDI, ZP, VES-GEM content, and physical appearance were analysed (Appendix A). Interestingly, minimal variations in size (~100 nm), ZP (~0 mV), PDI (~1), and VES-GEM levels (EE (%) > 90%) were reported after two months. Meanwhile, at 4 °C, the low temperature complicated the self-assembly phenomenon and the physical stability of micelles, and at 37 °C the chemical stability was thoroughly impacted by VES-GEM degradation, at RT stability was improved as micellization was favoured. Overall, the Pluronic^®^ F127/VES-GEM micelles exhibited better physical and chemical stability and were chosen as the most suitable candidate for subsequent studies.

### 3.8. Blood Compatibility

VES-GEM stock solutions in DMSO were diluted and mixed with fresh human blood, showing no haemolysis (Appendix A). Similarly, the Pluronic^®^/VES-GEM conjugate micelles showed no haemolytic activity, as attested by the transparency of the supernatants of each centrifuged aliquot, and further verified by the absorbance measurement (~0% haemolysis) (Figure 13A). These results add up to the suitability of both formulations for systemic delivery of VES-GEM.

### 3.9. Cell Viability Assay

The cytotoxic effects of Pluronic^®^ F68/VES-GEM (3/1), Pluronic^®^ F127/VES-GEM (1.5/1) and blank micelles, prepared according to Section 2.6, were assessed using BxPC3 cells, a rapidly proliferating and widely employed pancreatic cancer cell line with intermediate sensitivity to gemcitabine [86,87]. Free VES-GEM conjugate testing was omitted due to its precipitation in the culture medium, attributed to extremely low water solubility. The Pluronic^®^ F68/VES-GEM (3/1) micelles displayed higher efficacy in reducing cell viability below 50% compared to the negative control, achieving 48.49 ± 13.76% viability at a concentration of 25 µM. In contrast, the Pluronic^®^ F127/VES-GEM (1.5/1) micelles exhibited 43.38 ± 12.71% cell viability only at 100 µM (Figure 13B). Both Pluronic^®^ F127/VES-GEM (1.5/1) and Pluronic^®^ F68/VES-GEM (3/1) micelles demonstrated efficacy in reducing cell viability to less than 50%, beyond the performance of the free GEM.

The noteworthy performance of both the Pluronic^®^ F68/VES-GEM (3/1) and Pluronic^®^ F127/VES-GEM (1.5/1) micelles underscores their substantial capacity to inhibit cell viability compared to the free drug. This heightened efficacy is consistent with previous studies attributing enhanced drug delivery to the stability and encapsulation efficiency of Pluronic^®^ F68 and F127-based micelles [50,88]. The improved solubility profile of VES-GEM within the Pluronic^®^ F127 micelles likely contributes significantly to the observed cytotoxicity, emphasizing the pivotal role of adequate drug encapsulation and solubilization in augmenting therapeutic outcomes. Additionally, Pluronic^®^ F127/VES-GEM (1.5/1) displayed superior chemical and physical stability, as shown previously for T = 37 °C in which Pluronic^®^ F127/VES-GEM (1.5/1) were able to protect VES-GEM from degradation for longer time period when compared to Pluronic^®^ F68/VES-GEM (3/1), and displayed an improved physical stability profile by the absence of macroscopic aggregates and its ability to maintain size, PDI, and surface charge under an acceptable range. The increased core hydrophobicity conferred by the -PPO- segments of Pluronic^®^ F127 may help in improving VES-GEM encapsulation and its affinity, which may play a role in improving VES-GEM protection and subsequently its cell inhibition activity. The improved colloidal stability of the Pluronic^®^ F127/VES-GEM (1.5/1) micelles also aligned with their controlled release profile, in which moderate VES-GEM release after 72 h endows the system with interesting controlled release properties. At the same time, the micelles are still capable of accommodating VES-GEM and avoiding total release, acting as reservoirs. Conversely, Pluronic^®^ F68/VES-GEM (3/1) showed lower chemical and physicochemical stability profiles and the presence of visible aggregates, which suggested that the actual active encapsulated VES-GEM content is lower and, in this way, resulting in inferior cell viability reduction for the 100 µM concentration point.

### 3.10. Cell Uptake

In order to assess cellular internalization of the Pluronic^®^/VES-GEM micelles, cells were incubated with the dye-labelled formulations and later analysed by CLSM. Nile red, a model water-insoluble drug with strong red fluorescence, was used as a marker to study the intracellular localization of the prepared micelles. Both micelles showed the capability to be rapidly internalized by BxPC3 cells, and located near the nucleus (Figure 14 and Appendix A). Regardless of whether composed of Pluronic F68 or F127, the red fluorescence emitted by encapsulated Nile red was observed in the cell cytoplasm after 2 h of incubation. Moreover, the intranuclear presence of red fluorescence could reveal the delivery of cytoplasmatic release of Nile red and its further diffusion into the nucleus [89]. While the precise pathways for Nile red delivery within the Pluronic micelles remain elusive, these stabilized structures exhibit a notable ability to encapsulate hydrophobic molecules and facilitate their passage through the cell membrane [88]. Previous studies indicate that, while pure Nile red could enter cells through simple diffusion, block copolymer micelle uptake occurs via an endocytosis pathway [90]. Additionally, Pluronic micelles prominently interact with cellular membranes, enhancing both membrane microviscosity and permeability [57,91]. In summary, this interaction leads to an accelerated and more efficient internalization process.

## 4. Conclusions

In this work, vitamin E succinate-gemcitabine (VES-GEM) conjugate-loaded Pluronic^®^ F68 and Pluronic^®^ F127 micelles were explored as suitable delivery systems towards GEM delivery to pancreatic cancer. In order to increase the encapsulation efficiency of GEM in the micelles, a VES-GEM conjugate was prepared by conjugating hydrophobic VES at the 4-(N)-position of GEM, thereby assembling a multifunctional prodrug building block able to increase the lipophilicity of GEM and enable its encapsulation in micelles. Pluronic^®^ F68/VES-GEM (3/1) and Pluronic^®^ F127/VES-GEM (1.5/1) micelles were successfully prepared through the solvent evaporation method and characterized. Both formulations showed high encapsulation efficiency (>95%), a size of 100–150 nm, and slightly negative surface charge. Interestingly, VES-GEM solubilization was remarkably attained with Pluronic^®^ F68 and Pluronic^®^ F127 using smaller concentrations (<1% *w*/*v*) than those widely reported for other drugs and following an excipient reduction principle. The Pluronic F127^®^/VES-GEM conjugate micelles showed the best colloidal stability, attested by morphological and physicochemical stability in the first days, and by superior VES-GEM retention and protection, evidencing GEM protection inside the core in the VES-GEM form, as well as the best drug release profile, attaining a >50% cumulative release of VES-GEM after 72 h. In vitro cell viability showed that cells treated with the Pluronic^®^ F127/VES-GEM micelles could reach <50% cell viability for a concentration of 100 µM, and also showing the ability to preserve GEM activity by prodrug activation under endogenous stimuli, substantial cellular internalization, possibly aided by VES-GEM enhanced lipophilicity-mediated cell membrane entry. In addition, no haemolysis was detected for either formulation. Nevertheless, further studies elucidating the cell uptake mechanism and the specific in vivo stimuli-responsive features underlying GEM release, together with GEM quantification and pharmacokinetics, may engage in further exploration.

This work provides a new approach to enhancing GEM delivery to pancreatic cancer by increasing GEM solubility in amphiphilic block copolymers by means of a prodrug approach, hopefully paving the way for next-generation pancreatic cancer therapies.

## Figures and Tables

**Figure 1 pharmaceutics-16-00095-f001:**
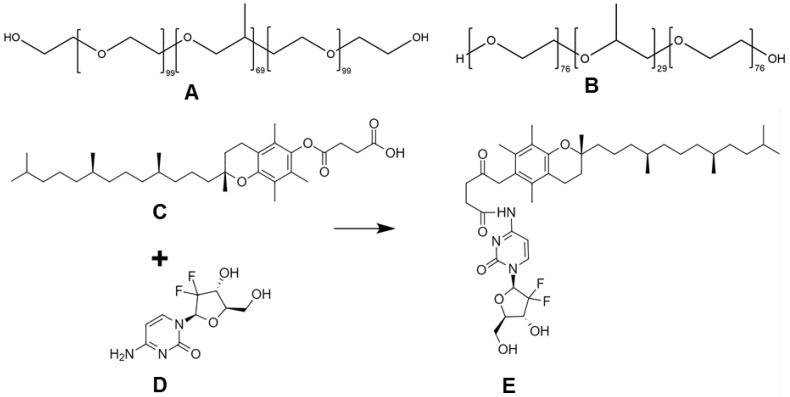
Chemical structures of Pluronic^®^ F127 (**A**), Pluronic^®^ F68 (**B**), the vitamin E succinate (**C**), gemcitabine (**D**), and the vitamin E-gemcitabine (VES-GEM) conjugate (**E**). VES is bridged with GEM through the formation of an amide bond established between the reactive amine and carboxylic acid groups of GEM and VES, respectively, yielding the VES-GEM conjugate.

**Figure 2 pharmaceutics-16-00095-f002:**
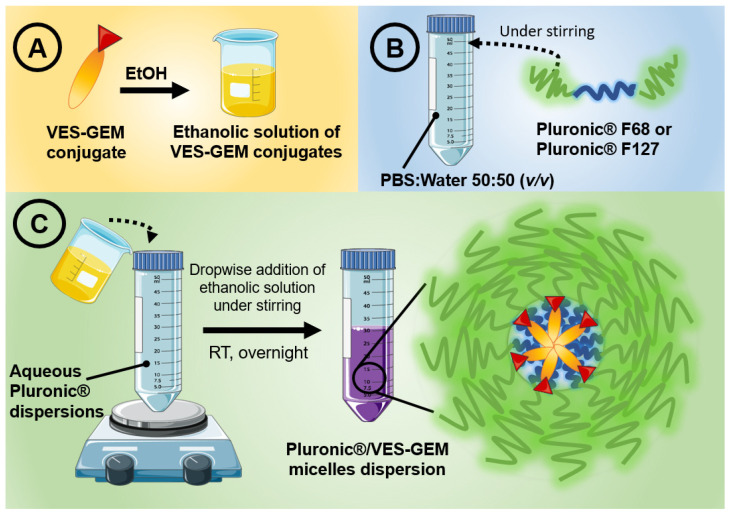
Schematic illustration of the preparation of Pluronic^®^/VES-GEM micelles using the dropwise addition and solvent evaporation method. (**A**) Solubilization of VES-GEM in ethanol (EtOH) with stirring, assembling an ethanolic solution of VES-GEM conjugates; (**B**) Dissolution of Pluronic^®^ F68 or Pluronic^®^ F127 in PBS:Water 50:50 (*v*/*v*); (**C**) Dropwise addition of ethanolic VES-GEM solution to the aqueous Pluronic^®^ solutions under stirring and ethanol evaporation overnight, assembling Pluronic^®^ F68/VES-GEM and Pluronic^®^ F127/VES-GEM micelles.

**Figure 3 pharmaceutics-16-00095-f003:**
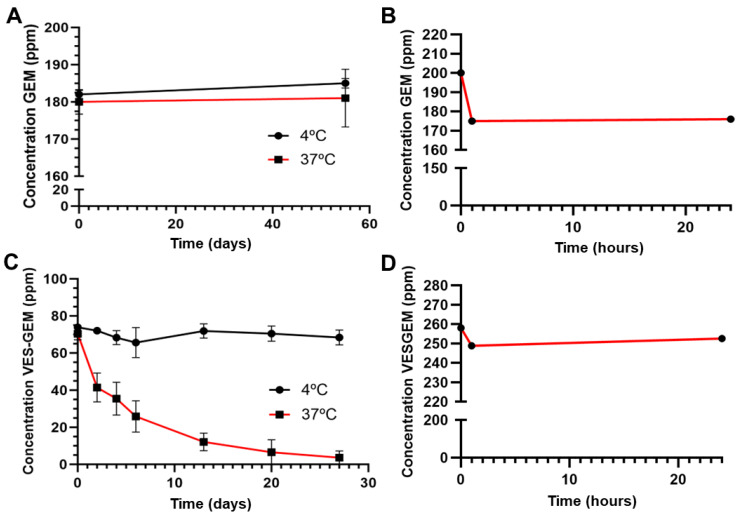
Stability of GEM in water (200 ppm) at 4 °C and 37 °C (**A**); the photostability of GEM at RT (**B**); the stability of VES-GEM in ethanol (100 ppm) (**C**) at 4 °C and 37 °C; and the VES-GEM conjugate photostability at RT (**D**).

**Figure 4 pharmaceutics-16-00095-f004:**
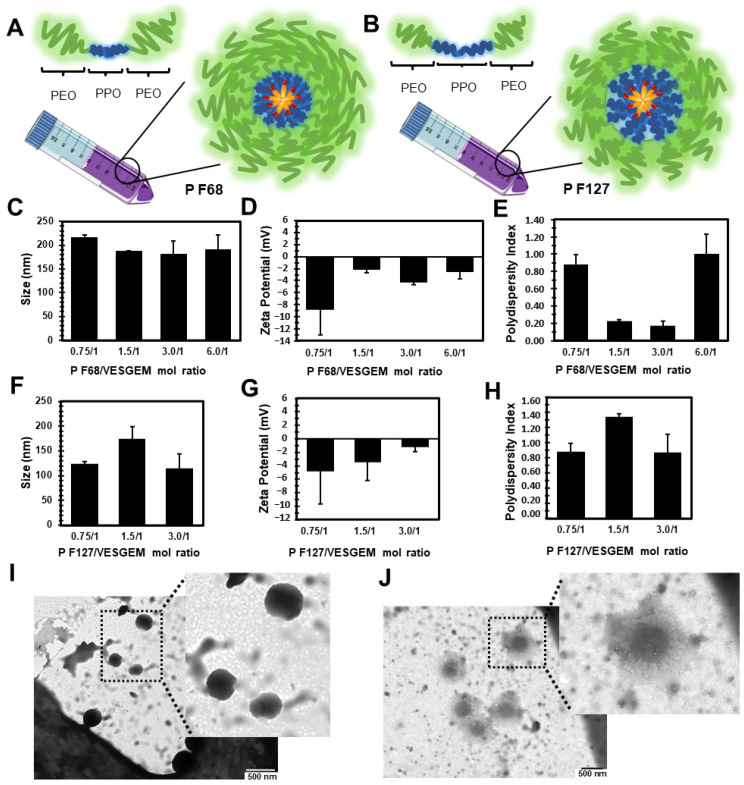
Schematic illustration of Pluronic^®^ F68, Pluronic^®^ F127 block copolymers, and Pluronic^®^/VES-GEM micelles (**A**,**B**). The molecular weight (length) of hydrophilic PEO chains and hydrophobic PPO chains depends on the type of Pluronic^®^ used, as Pluronic^®^ F127 has a more extensive hydrophobic PPO segment and a hydrophobic cavity more propense to accommodate the VES-GEM conjugate. Size (**C**), ZP (**D**), and PDI (**E**) of the Pluronic^®^ F68/VES-GEM conjugates micelles prepared with 0.75/1, 1.5/1, 3/1, and 6/1 molar ratios in PBS: water medium (50:50 *v*/*v*) and filtered. Size (**F**), ZP (**G**), and PDI (**H**) of the Pluronic^®^ F127/VES-GEM conjugates micelles prepared with 0.75/1, 1.5/1, and 3/1 molar ratios in PBS: water medium (50:50 *v*/*v*) and filtered. TEM pictures of Pluronic^®^ F68/VES-GEM (3/1) micelles (**I**) and Pluronic^®^ F127/VES-GEM (1.5/1) micelles (**J**). The scale bar is 500 nm.

**Figure 5 pharmaceutics-16-00095-f005:**
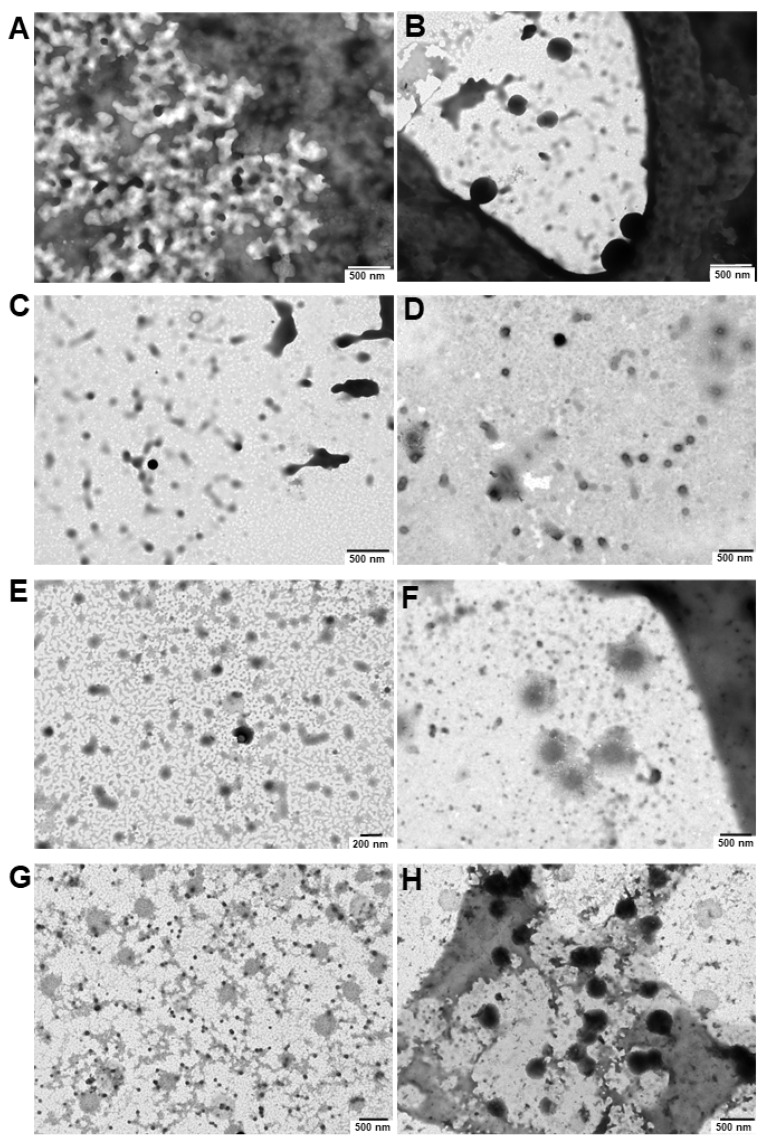
TEM pictures of the Pluronic^®^ F68/VES-GEM (3/1) micelles undiluted (**A**,**B**) and diluted 1:4 (**C**,**D**), and the Pluronic^®^ F127/VES-GEM (1.5/1) micelles undiluted (**E**,**F**) and diluted 1:4 (**G**,**H**). The photos were taken from two independently prepared samples of each formulation (**left** and **right**).

**Figure 6 pharmaceutics-16-00095-f006:**
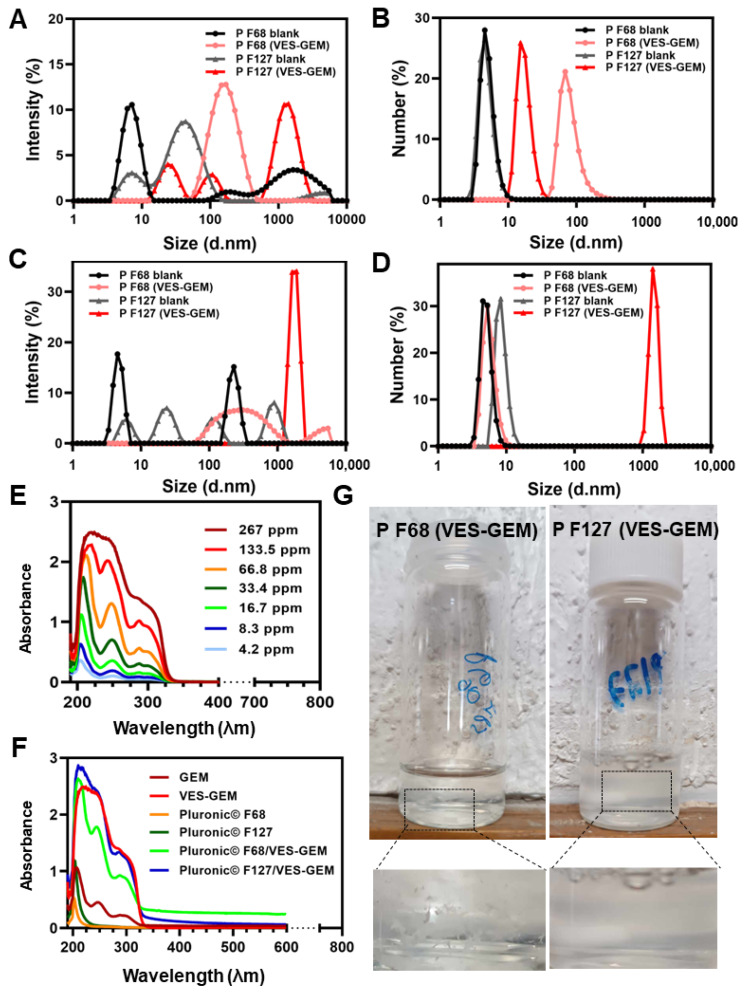
Intensity mode (%) (**A**) and number mode (%) (**B**) size distribution of the filtered Pluronic^®^ F68, Pluronic^®^ F68/VES-GEM (3/1), Pluronic^®^ F127, and Pluronic^®^ F127/VES-GEM (1.5/1) micelles in DLS. Intensity mode (%) (**C**) and number mode (%) (**D**) size distribution of the non-filtered Pluronic^®^ F68, Pluronic^®^ F68/VES-GEM (3/1), Pluronic^®^ F127, and Pluronic^®^ F127/VES-GEM (1.5/1) micelles in DLS. (**E**) UV-Vis absorption spectra of VES-GEM in ethanol, at RT, at different concentrations; (**F**) UV-Vis absorption spectra of the VES-GEM conjugate (in ethanol, at RT), Pluronic^®^ F68, Pluronic^®^ F68/VES-GEM (3/1), Pluronic^®^ F127, and Pluronic^®^ F127/VES-GEM (1.5/1) micelles in PBS: water 50:50 *v*/*v*, at RT; (**G**) the physical appearance of the Pluronic^®^ F68/VES-GEM (3/1)—**left**—and Pluronic^®^ F127/VES-GEM (1.5/1)—**right**—micelle dispersions. Abbreviations: P F68 blank—Pluronic^®^ F68 micelles; P F127 blank—Pluronic^®^ F127 blank micelles; P F68/VES-GEM—Pluronic^®^ F68/VES-GEM (3/1) micelles; P F127/VES-GEM—P F127—Pluronic^®^ F127/VES-GEM (1.5/1) micelles. For a better graphic visualization, a colour version of the article is advised.

**Figure 7 pharmaceutics-16-00095-f007:**
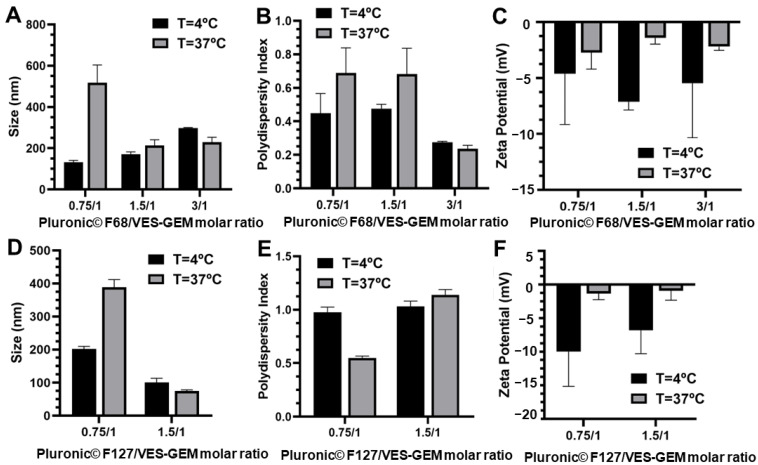
Influence of temperature on the size (**A**,**D**), PDI (**B**,**E**), and ZP (**C**,**F**) of the Pluronic^®^ F68/VES-GEM and Pluronic^®^ F127/VES-GEM conjugate micelles. The experiment was carried out at 4 °C and 37 °C.

**Figure 8 pharmaceutics-16-00095-f008:**
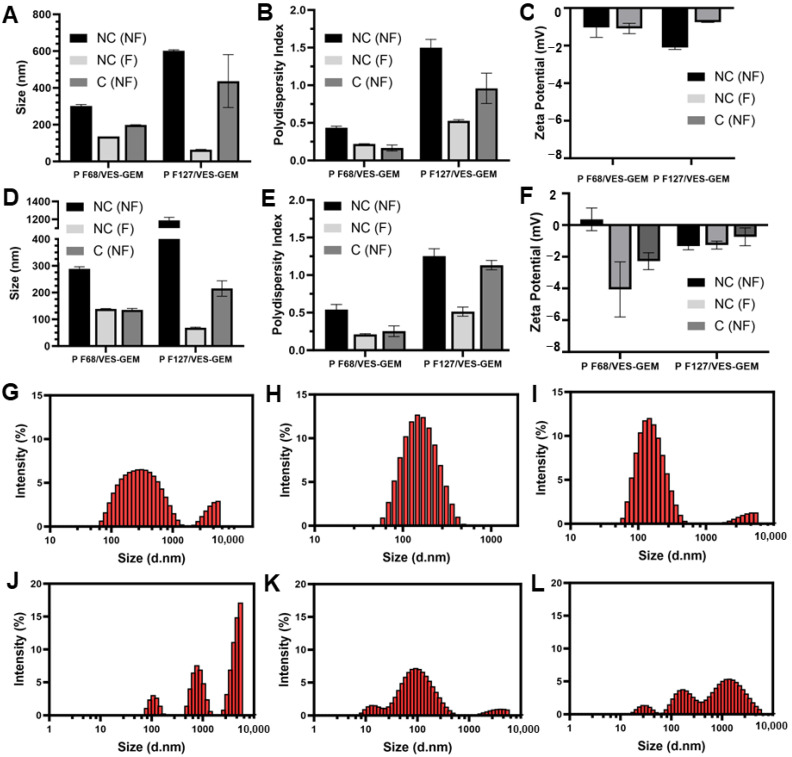
Influence of centrifugation as a purification method for preparing Pluronic^®^/VES-GEM micelles. Size (**A**), PDI (**B**), and ZP (**C**) of the Pluronic^®^ F68/VES-GEM (3/1) and Pluronic^®^ F127/VES-GEM (1.5/1) micelles, at 4000 rpm/30 min/T = 25 °C. Size (**D**), PDI (**E**), and ZP (**F**) of the Pluronic^®^ F68/VES-GEM (3/1) and Pluronic^®^ F127/VES-GEM micelles (1.5/1), at 12,000 rpm/20 min/T = 25 °C. Size distribution by intensity (%) of the non-centrifuged and non-filtered (**G**), non-centrifuged and filtered (**H**), and centrifuged, non-filtered (**I**) Pluronic^®^ F68/VES-GEM (3/1) micelles at 12,000 rpm/20 min/T = 25 °C. Size distribution by intensity (%) of the non-centrifuged and non-filtered (**J**), non-centrifuged and filtered (**K**), and centrifuged, non-filtered (**L**) Pluronic^®^ F127/VES-GEM (1.5/1) micelles at 12,000 rpm/20 min/T = 25 °C. Abbreviations: P F68 (VES-GEM)—Pluronic^®^ F68/VES-GEM (3/1) micelles; P F127 (VES-GEM)—Pluronic^®^ F127/VES-GEM (1.5/1) micelles; NC (NF)—non-centrifuged and non-filtered; NC (F)—non-centrifuged but filtered; C (NF) centrifuged, non-filtered.

**Figure 9 pharmaceutics-16-00095-f009:**
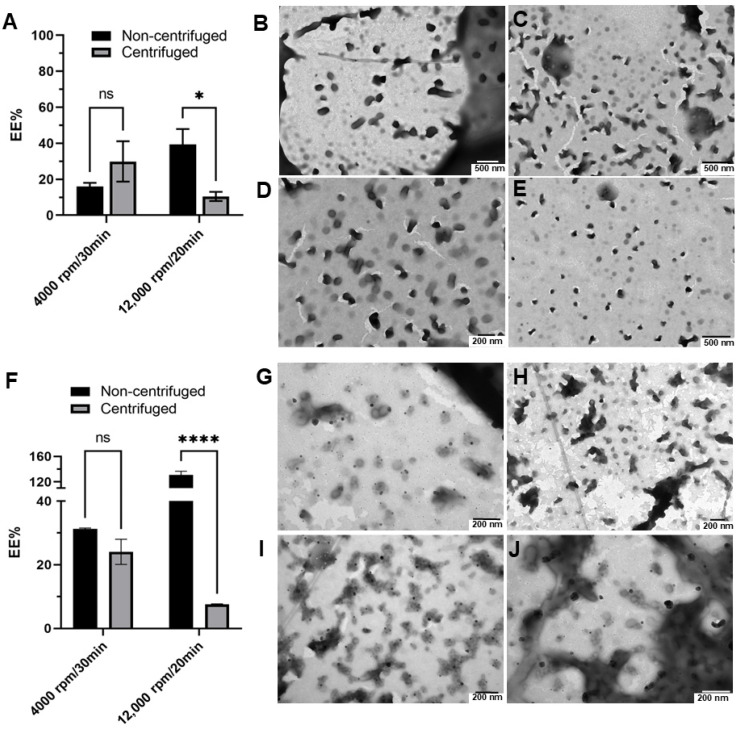
Effect of centrifugation on the encapsulation efficiency of thePluronic^®^ F68/VES-GEM (3/1) (**A**) and Pluronic^®^ F127/VES-GEM (1.5/1) conjugate micelles (**F**) [ns, non-significant * *p* < 0.05, **** *p* < 0.0001]. TEM picture of the Pluronic^®^ F68/VES-GEM (3/1) conjugate micelles (**B**–**E**) and the Pluronic^®^ F127/VES-GEM (1.5/1) conjugate micelles (**G**–**J**) after 4000 rpm/30 min/T = 25 °C centrifugation. Photos were taken from two independently prepared samples of each formulation (**left** and **right**).

**Figure 10 pharmaceutics-16-00095-f010:**
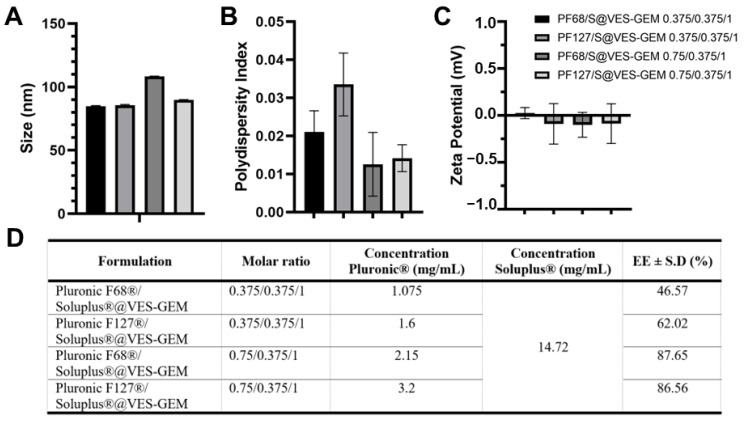
Size (**A**), PDI (**B**), ZP (**C**) of the Pluronic F68^®^/Soluplus^®^@VES-GEM (0.375/0.375/1 and 0.75/0.375/1) and Pluronic F127^®^/Soluplus^®^@VES-GEM (0.375/0.375/1 and 0.75/0.375/1) mixed micelles. (**D**) The encapsulation efficiency of the Pluronic^®^ F68/Soluplus^®^@VES-GEM and Pluronic^®^ F127/VES-GEM micelles. The Soluplus^®^ concentration was fixed at 14.72 mg/mL.

**Figure 11 pharmaceutics-16-00095-f011:**
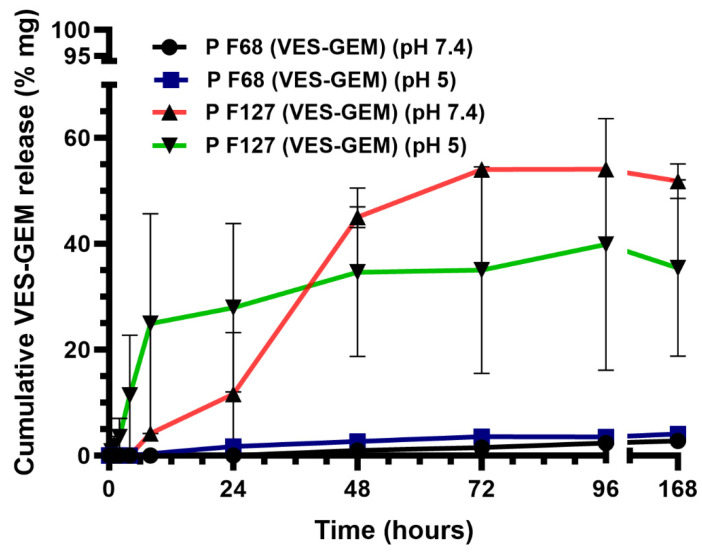
Drug release plots for the Pluronic^®^ F68/VES-GEM (3/1) and Pluronic^®^ F127/VES-GEM (1.5/1) micelles at pH 5 and pH 7.4. The formulations were incubated at 37 °C (100 rpm) and the medium was PBS supplemented with Tween 80 0.48% *v*/*v*. Abbreviations: P F68 (VES-GEM)—Pluronic^®^ F68/VES-GEM (3/1) micelles; P F127 (VES-GEM)—Pluronic^®^ F127/VES-GEM (1.5/1) micelles.

**Figure 12 pharmaceutics-16-00095-f012:**
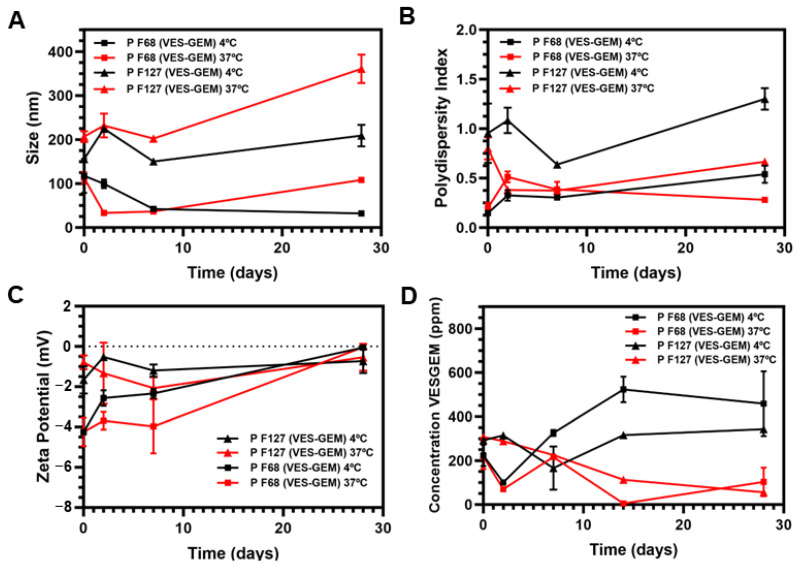
Stability evaluation of the Pluronic^®^ micelles through DLS and HPLC analysis at 4 °C and 37 °C. Size (**A**), PDI (**B**), ZP (**C**), and the VES-GEM content (**D**) of the Pluronic^®^ F68/VES-GEM (3/1) and Pluronic^®^ F127/VES-GEM (1.5/1) micelles. Abbreviations: P F68 (VES-GEM)—Pluronic^®^ F68/VES-GEM (3/1) micelles; P F127 (VES-GEM)—Pluronic^®^ F127/VES-GEM (1.5/1) micelles.

**Figure 13 pharmaceutics-16-00095-f013:**
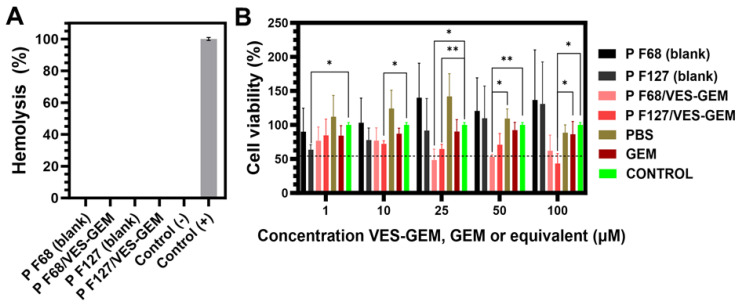
(**A**) Haemolytic activity for Pluronic^®^ and Pluronic^®^/VES-GEM conjugate micelles in PBS: water 50:50 *v*/*v*. (**B**) In vitro cytotoxicity expressed as cell viability (%) of various concentrations (1 μM, 10 μM, 25 μM, 50 μM, 100 μM) of formulations (blank Pluronic^®^ F68 and Pluronic^®^ F127 micelles, Pluronic^®^ F68/VES-GEM (3/1) and Pluronic^®^ F127/VES-GEM (1.5/1) micelles) using Quant-iT™ *PicoGreen*™ (**B**) assays. * *p* < 0.05, ** *p* < 0.01. Abbreviations: P F68—Pluronic^®^ F68 micelles; P F68 (VES-GEM)—Pluronic^®^ F68/VES-GEM (3/1) micelles; P F127—Pluronic^®^ F127 micelles; P F127 (VES-GEM)—Pluronic^®^ F127/VES-GEM (1.5/1) micelles; GEM—gemcitabine.

**Figure 14 pharmaceutics-16-00095-f014:**
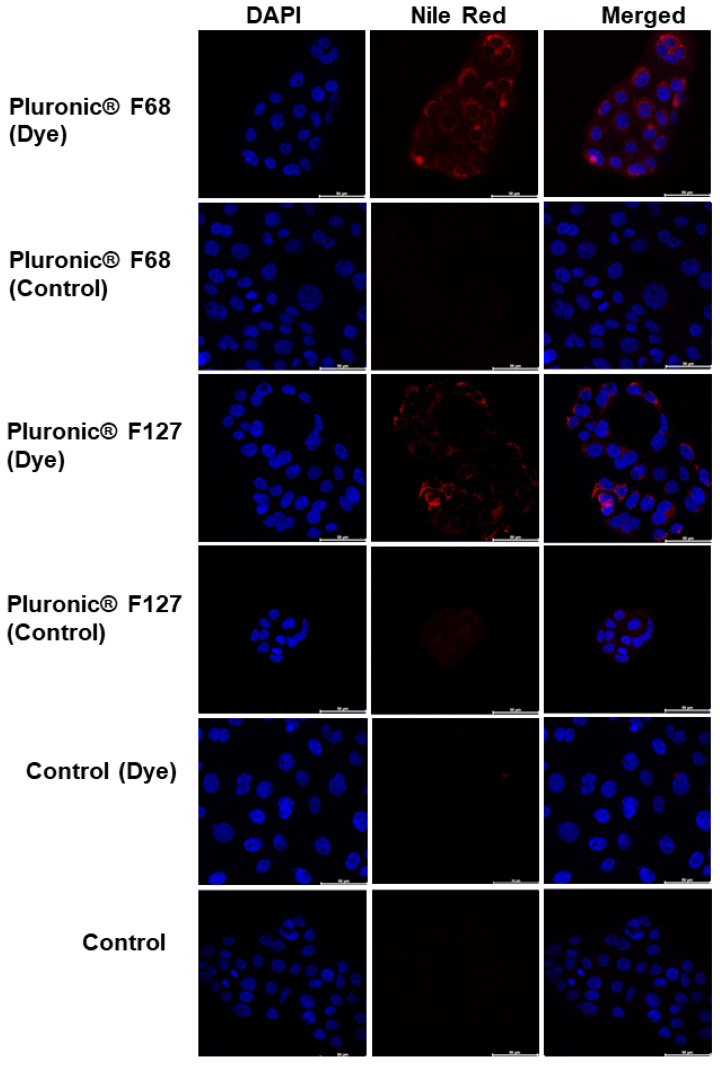
Intracellular uptake of Nile Red-loaded Pluronic^®^ F68/VES-GEM (3/1) and Pluronic^®^ F127/VES-GEM (1.5/1) micelles. Cell nuclei were stained with DAPI in blue. No obvious red fluorescence signal (Nile red) was observed in the non-loaded micelles and the control cells treated with free Nile Red. The scale bar is 50 µm.

**Table 1 pharmaceutics-16-00095-t001:** Composition of Pluronic^®^/VES-GEM micelles.

Formulations	Polymer-to-Conjugate Molar Ratio	% *w*/*v* of Pluronic^®^	% *w*/*v* of VES-GEM	% *w*/*w* VES-GEM in Pluronic^®^	Concentration of Pluronic^®^ (mg/mL)	Concentration of Pluronic^®^ (mM)	Concentration of VES-GEM (mg/mL)
Pluronic^®^ F68/VES-GEM	0.75/1	0.22	0.027	12.40	2.15	0.256	0.267
	1.5/1	0.43		6.20	4.30	0.512	
	3/1	0.86		3.10	8.60	1.024	
	6/1	1.72		1.55	17.21	2.048	
Pluronic^®^ F127/VES-GEM	0.75/1	0.32	0.027	8.18	3.23	0.258	
	1.5/1	0.65		4.13	6.45	0.517	
	3/1	1.30		2.07	12.90	1.034	

**Table 2 pharmaceutics-16-00095-t002:** Prediction of several chemical parameters for the VES-GEM conjugate, using the Chemicalize and MarvinView software models.

Parameter	Result
Molecular weight	775.976
Isotope formula	C_42_H_63_F_2_N_3_O_8_
Isoelectric point	6.26
LogP	8.93
LogD	8.93
HLB (Chemaxon)	9.53
HLB (Davies)	10.13
HLB	8.64
Intrinsic solubility	−10.76 (logS)
Solubility at pH 7.4	−10.76 (logS)
Solubility category	Low (lower than 0.01 mg/mL)
Predominant species (pH 4–9)	Non-ionized
Charge (pH 4–9)	~0
H bond donor/acceptor sites	3/15

**Table 3 pharmaceutics-16-00095-t003:** Size, ZP, and PDI of the filtered and non-filtered Pluronic^®^ F68, Pluronic^®^ F68/VES-GEM, Pluronic^®^ F127, and Pluronic^®^ F127/VES-GEM micelles.

Formulations	Molar Ratio	Filtered	Size (nm) ± S.D.	ZP (mV) ± S.D.	PDI ± S.D.
Pluronic^®^ F68	-	Yes	34.24 ± 13.11	−3.51 ± 0.88	0.486 ± 0.039
Pluronic^®^ F68	-	No	185.73 ± 43.16	−4.33 ± 1.11	0.350 ± 0.024
Pluronic^®^ F68/VES-GEM	3/1	Yes	140.01 ± 0.28	−5.38 ± 1.60	0.213 ± 0.011
Pluronic^®^ F68/VES-GEM	3/1	No	284.27 ± 1.04	0.04 ± 0.83	0.497 ± 0.074
Pluronic^®^ F127	-	Yes	25.57 ± 0.47	−2.14 ± 0.14	0.493 ± 0.086
Pluronic^®^ F127	-	No	307.57 ± 49.54	−9.11 ± 0.62	0.456 ± 0.025
Pluronic^®^ F127/VES-GEM	1.5/1	Yes	136.67 ± 25.70	−1.54 ± 1.08	1.234 ± 0.021
Pluronic^®^ F127/VES-GEM	1.5/1	No	3282.33 ± 641.42	−0.69 ± 0.28	0.477 ± 0.340

**Table 4 pharmaceutics-16-00095-t004:** Size and PDI of filtered and non-filtered Pluronic^®^ F127/VES-GEM (1.5/1) subjected to sonication at RT.

Formulations	Molar Ratio	Sonication Time	Filtered before Measurement	Size (nm) ± S.D.	PDI ± S.D.
Pluronic^®^ F127	-	0 min	Yes	35.82 ± 0.50	0.536 ± 0.030
			No	100.47 ± 24.63	0.359 ± 0.107
		5 min	Yes	23.30 ± 0.54	0.435 ± 0.011
			No	270.03 ± 69.35	0.379 ± 0.031
		10 min	Yes	31.65 ± 2.98	0.421 ± 0.052
			No	353.4 ± 45.8	0.380 ± 0.035
		30 min	Yes	33.16 ± 5.26	0.489 ± 0.060
			No	379.33 ± 40.86	0.400 ± 0.025
		90 min	Yes	28.62 ± 0.21	0.419 ± 0.005
			No	94.07 ± 13.84	0.378 ± 0.038
Pluronic^®^ F127/VES-GEM	1.5/1	0 min	Yes	131.33 ± 23.06	0.973 ± 0.083
			No	2146.33 ± 276.06	0.945 ± 0.184
		5 min	Yes	111.27 ± 6.77	0.928 ± 0.011
			No	2501.3 ± 117.3	0.827 ± 0.161
		10 min	Yes	103.36 ± 17.04	0.814 ± 0.039
			No	2265.33 ± 147.82	0.589 ± 0.247
		30 min	Yes	227.80 ± 76.42	0.568 ± 0.194
			No	2472.67 ± 267.82	0.640 ± 0.107
		90 min	Yes	66.45 ± 3.66	0.717 ± 0.012
			No	2519.67 ± 298.92	0.617 ± 0.100

**Table 5 pharmaceutics-16-00095-t005:** Size and PDI of filtered and non-filtered Pluronic^®^ F127/VES-GEM (1.5/1) subjected to ultrasonication at RT.

Formulations	Molar Ratio	Ultrasonication Time	Filtered before Measurement	Size (nm) ± S.D.	PDI ± S.D.
Pluronic^®^ F127	-	0 s	Yes	35.82 ± 0.50	0.536 ± 0.030
			No	100.47 ± 24.63	0.359 ± 0.107
		3 s	Yes	36.03 ± 0.93	0.435 ± 0.011
			No	37.47 ± 6.27	0.402 ± 0.112
		6 s	Yes	31.89 ± 0.44	0.576 ± 0.047
			No	282.27 ± 176.53	0.683 ± 0.055
		12 s	Yes	39.93 ± 0.10	0.782 ± 0.006
			No	408.57 ± 84.89	0.556 ± 0.088
		24 s (6 s on, 3 s off)	Yes	46.09 ± 1.45	0.763 ± 0.027
			No	91.21 ± 21.43	0.559 ± 0.127
		36 s (6 s on, 3 s off)	Yes	41.09 ± 1.03	0.883 ± 0.029
			No	265.13 ± 109.23	0.623 ± 0.188
Pluronic^®^ F127/VES-GEM	1.5/1	0 s	Yes	131.33 ± 23.06	0.973 ± 0.083
			No	2146.33 ± 276.06	0.945 ± 0.184
		3 s	Yes	146.9 ± 1.45	0.446 ± 0.033
			No	863.97 ± 54.17	0.582 ± 0.026
		6 s	Yes	158.67 ± 6.84	0.400 ± 0.059
			No	589.07 ± 10.44	0.541 ± 0.037
		12 s	Yes	169.00 ± 10.77	0.402 ± 0.015
			No	506.23 ± 33.59	0.551 ± 0.045
		24 s (6 s on, 3 s off)	Yes	407.80 ± 7.80	0.310 ± 0.022
			No	307.57 ± 5.46	0.461 ± 0.033
		36 s (6 s on, 3 s off)	Yes	186.50 ± 4.78	0.294 ± 0.013
			No	395.83 ± 17.97	0.443 ± 0.053

## Data Availability

Datasets used and analysed during the current study are available from the corresponding author upon reasonable request.

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
