# Peer review of "Gemcitabine-Vitamin E Prodrug-Loaded Micelles for Pancreatic Cancer Therapy"

_pharmaceutics, 2024, doi:10.3390/pharmaceutics16010095_

Round 1

Reviewer 1 Report

Comments and Suggestions for Authors

1. VES-GEM can self-assemble by itself. How do you confirm? How to purify the unencapsulated VES-GEM?

2. The modification site is amino group of GEM. But the amino group is very important for activity of GEM. Some evidence should be supplied on the activation of VES-GEM.

3. It is difficult to imagine that VES-GEM encapsulated in the micelle shows higher toxicity than free GEM. Especially, VES-GEM is an in-activated state. Note that usually small molecule drug should be more toxic than its nanoformulation. The authors should discuss more on this point. 

4. Control group, VES-GEM, should be included in some experiments, such as size and cytotoxicity.

5. Please give some general discussion on the advantages and functions of nanomedicine in the introduction, such as long circulation, improved targeting ability, enhanced cellular uptake (https://doi.org/10.1016/j.addr.2023.114895; https://doi.org/10.1021/jacs.0c09029).

6. More research background on  Pluronic micelle should be introduced in the introduction (https://doi.org/10.3390/molecules26123610; https://doi.org/10.3390/polym15102249; https://doi.org/10.3390/jfb9010011).

Author Response

The authors appreciate the careful review and greatly acknowledge the comments that the Reviewers have provided on our manuscript. We have carefully revised the manuscript and have made the recommended changes and answered in detail to the questions raised. Additional information was also added when appropriate. All changes made to the text are highlighted in yellow colour. Please find below a point-by-point list of answers to the reviewers’ concerns.

REVIEWER #1

  1. VES-GEM can self-assemble by itself. How do you confirm? How to purify the unencapsulated VES-GEM?

ANSWER: VES-GEM is an amphiphilic conjugate in which vitamin E portion corresponds to the hydrophobic part, and GEM to the hydrophilic head. It is possible VES-GEM may self-assemble into micelles as other amphiphilic building blocks. In regard to lipidic prodrugs, self-assemble may form other structures than micelles (e.g. squalenoyl-GEM structures), also called, generically, nanoassemblies. However, such self-assembly properties are not common and only some conjugates may be chemically predisposed to form such nanostructures. The ability to self-assemble into nanoassemblies has been studied elsewhere and it seems hydrophilic–lipophilic balance, HLB, plays a key role in terms of predictive potential (https://doi.org/10.1021/acs.bioconjchem.1c00051). In this study, HLB below 8.34 for the lipidic prodrug conjugate was required to self-assemble into nanoparticles. VES-GEM has higher HLB, which may indicate unfavorable potential towards self-assembly. Moreover, considering direct dissolution method in which the majority of micelle building blocks are able to self-assemble, the conjugate is extremely hydrophobic, and has extremely low water solubility, which means even if some micelles or nanostructures may be formed through direct dissolution in water, the yield would be extremely low. When considering solvent evaporation method, the one employed in this work, the conjugate is expected to be encapsulated in its vast majority into micelles. However, it is possible some conjugate may self-assemble into nanoparticles, which can be only residual as we expect encapsulation in the hydrophobic core of micelles would be favourable when compared to VES-GEM units self-assembly. Nonetheless, nanoparticles around 300 nm and PDI~0.3 could be obtained recurring to the same solvent evaporation methodology, in which VES-GEM was dissolved in ethanol in absence of any surfactant. Overall, whole VES-GEM nanoparticles may be obtained when no surfactant is present, but the size would be ~ 300 nm and PDI above 0.3 (after filtration) and in presence of large aggregates. A paragraph was introduced on page 18 (lines 676-681) explaining VES-GEM nanoparticles production: “The ability of VES-GEM to self-assemble in nanostructures was also explored following the same method and inspired by previous studies which reported nanostructures made of lipid-GEM conjugates without additional surfactant addition [14,27]. The formulation showed large aggregates and particle size of 286 ± 40.26 nm, ZP of -0.4. ± 0.75 mV and PDI= 0.36 ± 0.14 and a yield below 10% due to the low solubility of the low water solubility, and was not further explored.”

According to the extremely low water solubility of VES-GEM and the presence of amphiphilic block copolymers able to accommodate VES-GEM in their hydrophobic core, it is expected most VES-GEM would be in encapsulated form, also attested by encapsulation efficiency measurements. However, direct quantification of VES-GEM may produce erroneous results, as residual undissolved VES-GEM may be solubilized in ethanol and then giving an illusion of extremely high encapsulation efficiency. Hence, purification methods were undertaken to purify the of Pluronic®/VES-GEM micelles regarding 1) elimination of free VES-GEM, 2) elimination of polymers, 3) elimination of aggregates, namely purification by centrifugation and by filtration, as reported on page 18 (lines 655-669) regarding filtration and on page 24 (lines 755-780), for centrifugation. Although these methods of purification have been broadly employed to purify micelle systems, it is not clear whether they may destabilize the system and retain or precipitate the desired structures giving rise to incorrect measurements. Dialysis was not explored as purification method as micelles are very dynamic colloidal systems and this procedure may destabilize the system and contribute to drug leakage. Hence, the authors have chosen to proceed with the system with the minimum number of external variables that may impact overall composition and stability.

  1. The modification site is amino group of GEM. But the amino group is very important for activity of GEM. Some evidence should be supplied on the activation of VES-GEM.

ANSWER: The authors would like to thank the reviewer’s comment. Several studies have reported GEM coupling to lipidic moieties through an amide bond at 4-N-position of GEM, namely squalenoyl-GEM (https://doi.org/10.1016/j.ejpb.2019.09.017; https://doi.org/10.3390/cancers12071895; https://doi.org/10.1039/c7tb02899g); N-octadecanoyl-GEM (http://dx.doi.org/10.1016/j.ijpharm.2012.03.046); solanesol-glutaric acid-GEM (https://doi.org/10.1021/acs.bioconjchem.1c00051); cholesterol-glutaric acid-GEM (https://doi.org/10.1021/acs.bioconjchem.1c00051); reduced squalenic acid-GEM (https://doi.org/10.1021/acs.bioconjchem.1c00051); vitamin E-glutaric acid-GEM (https://doi.org/10.1021/acs.bioconjchem.1c00051), C14-GEM (https://doi.org/10.1021/acsnano.2c07861), linoleic acid-GEM (https://doi.org/10.1021/acsami.9b16209), retinoic acid-GEM (https://doi.org/10.1016/j.msec.2020.111251). The conjugation of hydrophobic derivatives to 4-N position of GEM and not to other positions has to do with GEM protection to deamination processes, simplified chemistry and possibility to form ideally self-assembled structures and possibility to explore enhanced permeability and retention effect. Encapsulation in micelles of the amine-protected GEM, such as in our case (VES-GEM) is expected to maximize protection of GEM, both chemically and in pharmacokinetic terms (https://doi.org/10.1016/j.msec.2020.111251; https://doi.org/10.1021/mp4005904; https://doi.org/10.1016/j.ijpharm.2014.04.021; https://doi.org/10.1039/C7RA02909H).

Activation of VES-GEM is expected to occur under cleavage of amide bond by cathepsin B to a certain degree (i.e. the cleavage may be incomplete), which has been reported extensively in literature to similar GEM conjugates (https://doi.org/10.1039/C4RA13870H; https://doi.org/10.1021/acsnano.2c07861; https://doi.org/10.1016/j.addr.2010.12.005; https://doi.org/10.1016/j.addr.2019.01.010), while other enzymes may help releasing GEM, such as amidases, phospholipases (doi:10.1016/j.ejpb.2021.05.018), esterases (https://doi.org/10.1002/cplu.202000253), and hydrolysis (https://doi.org/10.1002/smll.202107712) to name a few. This information is explained on page 3 (lines 100-107).

Additional studies show GEM modification in the amine group did not impair anticancer activity of GEM prodrugs when compared to free GEM (https://doi.org/10.1039/C8BM00946E; https://doi.org/10.1021/acsmacrolett.7b00160;https://doi.org/10.1016/j.jconrel.2015.01.021).

  1. It is difficult to imagine that VES-GEM encapsulated in the micelle shows higher toxicity than free GEM. Especially, VES-GEM is an in-activated state. Note that usually small molecule drug should be more toxic than its nanoformulation. The authors should discuss more on this point.

ANSWER: The authors agree with the reviewer’s comment. While it can be expected the activated “free” drug (GEM) can exert increased anticancer effect when compared to its prodrug derivative, several aspects should be taken into consideration that may explain similar or even increase in therapeutic action of VES-GEM, namely increased cell uptake of GEM in VES-GEM form due to the conjugate’s hydrophobicity, thus favouring membrane passage (this may be expected in less extension, as the micelles already can undergo uptake to intracellular milieu); maximized release of VES-GEM (and then GEM) intracellularly via micelles, which in the case of free GEM would be more difficult as GEM is hydrophilic, multidrug resistant cells may hinder GEM entry through its specific transporters and GEM is rapidly inactivated and cleared when in contact with biological environment. Additionally, the lipidic block may also exert anticancer action, which may add up to the overall anticancer effect shown. GEM action is expected to occur after prodrug activation (comment #2). Several studies have reported GEM hydrophobic prodrug displaying increased anticancer potential when compared to free GEM, such as squalenoyl-GEM (doi:10.1016/j.nano.2011.02.012; https://doi.org/10.1016/j.bbadis.2022.166614; https://doi.org/10.1021/acs.bioconjchem.1c00051), linoleic acid-GEM (https://doi.org/10.1021/acsami.9b16209);

  1. Control group, VES-GEM, should be included in some experiments, such as size and cytotoxicity.

ANSWER: The authors appreciate the reviewer’s suggestion. According to comment #1, an explanation was provided on page 18 (lines 677-682) as requested by the reviewer. Regarding in vitro cytotoxicity studies, free VES-GEM was not included due to its extremely low aqueous solubility and precipitation in medium even for concentration below 1 µM. An explanation was provided on page 34 (lines 1031-1033): “Free VES-GEM conjugate testing was omitted due to its precipitation in culture medium, attributed to extremely low water solubility.”

  1. Please give some general discussion on the advantages and functions of nanomedicine in the introduction, such as long circulation, improved targeting ability, enhanced cellular uptake (https://doi.org/10.1016/j.addr.2023.114895; https://doi.org/10.1021/jacs.0c09029).

ANSWER: The authors would like to thank the reviewer’s suggestion. Additional information was added as requested on pages 2-3 (lines 86-92): “Stability and blood circulation improvements may be attributed to the stealth features of PEGylated nanosystems which can help maximizing blood circulation profile of drugs and decrease aggregation [10]. Additional stimuli-responsive attributes and advanced targeting moieties can be included in nanosystems to improve tumor targeting and enable selective drug release, in accordance to the natural and pathophysiological barriers [11].”

  1. More research background on Pluronic micelle should be introduced in the introduction (https://doi.org/10.3390/molecules26123610; https://doi.org/10.3390/polym15102249; https://doi.org/10.3390/jfb9010011).

ANSWER: The authors would like to acknowledge the reviewer’s comment. Additional information was added as requested on pages 3-4 (lines 125, 127, 134-137): “ In addition, Pluronic® F68 and Pluronic® F127 display physical state of flake – F – high hydrophilicity, solubility and biocompatibility, the hydrophilic-lipophilic balance (HLB) value is within 20-29 and PPO chain length in Pluronic® F127 is twice as the one in Pluronic® F68 [48,49].”

Reviewer 2 Report

Comments and Suggestions for Authors

In this study, Pluronic® F68 and F127 micelles loaded with the vitamin E succinate-gemcitabine (VES-GEM) conjugatate were investigated as viable delivery vehicles for GEM delivery to pancreatic cancer. Both formulations exhibited 100-150nm micelle size, high GEM encapsulation efficiency (>95%), and slightly negative surface charge. Pluronic F127/VES-GEM conjugate micelles showed the best colloidal stability, attested by morphological and physicochemical stability in the first days, and by superior VES-GEM retention and protection, evidencing GEM protection inside the core in VES-GEM form. No haemolysis was detected for both formulations. 

The work is deserving of publication due to its novelty and the volume of material presented. However, some modifications are necessary.

The scale bar are not visible in Fig. 5 and 9, please correct. There are 2 TEMs in Figures 5A, Figures 5B, Figures 5C, Figures 9B, Figures 9D. This is not explained in any way in the text. Please provide a detailed description of the TEM pictures in the text and label each image with a separate letter.

Fig. 14 needs clear scale bars.

DLS distributions (Fig.6 A,B,C) looks very confusing. The distribution contains numerous peaks and therefore cannot be analysed easily.

 Figure 8 is missing some information. What is highlighted in color in Fig.8 A,B,D,E?

Author Response

The authors appreciate the careful review and greatly acknowledge the comments that the Reviewers have provided on our manuscript. We have carefully revised the manuscript and have made the recommended changes and answered in detail to the questions raised. Additional information was also added when appropriate. All changes made to the text are highlighted in yellow colour. Please find below a point-by-point list of answers to the reviewers’ concerns.

REVIEWER #2

In this study, Pluronic® F68 and F127 micelles loaded with the vitamin E succinate-gemcitabine (VES-GEM) conjugatate were investigated as viable delivery vehicles for GEM delivery to pancreatic cancer. Both formulations exhibited 100-150nm micelle size, high GEM encapsulation efficiency (>95%), and slightly negative surface charge. Pluronic F127/VES-GEM conjugate micelles showed the best colloidal stability, attested by morphological and physicochemical stability in the first days, and by superior VES-GEM retention and protection, evidencing GEM protection inside the core in VES-GEM form. No haemolysis was detected for both formulations. 

The work is deserving of publication due to its novelty and the volume of material presented. However, some modifications are necessary.

ANSWER: The authors would like to thank the reviewer for the positive observations and insights.

The scale bar are not visible in Fig. 5 and 9, please correct. There are 2 TEMs in Figures 5A, Figures 5B, Figures 5C, Figures 9B, Figures 9D. This is not explained in any way in the text. Please provide a detailed description of the TEM pictures in the text and label each image with a separate letter.

ANSWER: The authors apologize for these edition issues. The scale bar was corrected, TEM images were labelled with a letter and correctly inserted in text, highlighted in yellow. Photos were taken from two independently prepared samples of each formulation (left and right), as explained now in the caption.

Fig. 14 needs clear scale bars.

ANSWER: The authors would like to acknowledge the reviewer’s comment. The scale bars were improved as per the reviewer’s suggestion.

DLS distributions (Fig.6 A,B,C) looks very confusing. The distribution contains numerous peaks and therefore cannot be analysed easily.

ANSWER: The authors appreciate the reviewer’s comment. In order to save space, the number of plots was maintained, but a symbol code was included to help the readers identify each peak.

 Figure 8 is missing some information. What is highlighted in color in Fig.8 A,B,D,E?

ANSWER: The authors thank the reviewer’s comment. The caption is the same for all A to F plots and it is now explained in the caption as follows “NC (NF) - non-centrifuged and non-filtered; NC (F) - non-centrifuged but filtered; C (NF) centrifuged, non-filtered.”

Round 2

Reviewer 1 Report

Comments and Suggestions for Authors

The authors addressed the concerns well.